

# Reactive nitrogen fluxes over peatland and forest ecosystems using micrometeorological measurement techniques

Christian Brümmer[1], Jeremy J. Rüffer[1], Jean-Pierre Delorme[1], Pascal Wintjen[1], Frederik Schrader[1], Burkhard Beudert[2], Martijn Schaap[3,4], Christof Ammann[5]

[1]Thünen Institute of Climate-Smart Agriculture, 38116 Braunschweig, Germany
[2]Bavarian Forest National Park, 94481, Grafenau, Germany
[3]TNO, Department of Climate, Air and Sustainability, Utrecht, NL-3584, The Netherlands
[4]Institute of Meteorology, Freie Universität Berlin, 12165 Berlin, Germany
[5]Climate and Agriculture Group, Agroscope, Reckenholzstrasse 191, 8046, Zürich, Switzerland

*Correspondence to*: Christian Brümmer (christian.bruemmer@thuenen.de)

**Abstract.** Interactions of reactive nitrogen ($N_r$) compounds between the atmosphere and the earth's surface play a key role in atmospheric chemistry and in understanding nutrient cycling of terrestrial ecosystems. While continuous observations of inert greenhouse gases through micrometeorological flux measurements have become a common procedure, information about temporal dynamics and longer-term budgets of $N_r$ compounds is still extremely limited. Within the framework of the research

projects NITROSPHERE and FORESTFLUX, field campaigns were carried out to investigate the biosphere-atmosphere exchange of selected $N_r$ compounds over different land surfaces. The aim of the campaigns was to test and establish novel measurement techniques in eddy-covariance setups for continuous determination of surface fluxes of ammonia ($NH_3$) and total reactive nitrogen ($\Sigma N_r$) using two different analytical devices. While high-frequency measurements of $NH_3$ were conducted with a quantum cascade laser (QCL) absorption spectrometer, a custom-built converter called Total Reactive Atmospheric

Nitrogen Converter (TRANC) connected and operated upstream of a chemiluminescence detector (CLD) was used for the measurement of $\Sigma N_r$. As high-resolution data of $N_r$ surface-atmosphere exchange are still scarce, but highly desired for testing and validating local inferential and larger scale models, we provide access to campaign data including concentrations, fluxes, and ancillary measurements of meteorological parameters. Campaigns (*n*=4) were carried out in natural (forest) and semi-natural (peatland) ecosystem types. The published datasets stress the importance of recent advancements in laser spectrometry

and help improve our understanding of the temporal variability of surface-atmosphere exchange in different ecosystems, thereby providing validation opportunities for inferential models simulating the exchange of reactive nitrogen. The dataset has been placed in the Zenodo repository (http://doi.org/10.5281/zenodo.4513854; Brümmer et al., 2021) and contains individual data files for each campaign.



## 1 Introduction

The term 'reactive nitrogen' ($N_r$) describes all forms of nitrogen that are biologically, photochemically, and radiatively active. The most important substances are nitric oxide (NO), nitrogen dioxide ($NO_2$), nitrate ($NO_3^-$), nitric acid ($HNO_3$), ammonium ($NH_4^+$), and ammonia ($NH_3$). Historically, their low availability had been the main limiting factor for the productivity of natural ecosystems. However, with the invention of the Haber-Bosch process in 1908, inexpensive industrial mass production of synthetic $N_r$ fertilizers through synthesizing $NH_3$ from di-nitrogen ($N_2$) and hydrogen became viable, thereby inducing a

number of positive and unintended negative consequences (Erisman et al., 2008; Sutton et al., 2011). A clear benefit was the stabilization of the food supply for populations in developed countries, whereas substantial $N_r$ losses into the environment caused a number of adverse effects for terrestrial and aquatic ecosystems. These were for example impacts on greenhouse gas exchange such as enhanced nitrous oxide ($N_2O$) and $NH_3$ emissions and inhibition of soil methane ($CH_4$) oxidation. Other effects comprised direct toxicity of plants followed by biodiversity loss, while acidification and eutrophication of soils and

inshore waters, respectively, were further observed (Galloway et al., 2003; Flechard et al., 2011).

Biosphere-atmosphere interactions of $N_r$ compounds play a key role in atmospheric chemistry and in nutrient dynamics of terrestrial and aquatic ecosystems (Ollinger et al., 2002; Farmer et al., 2008). While cropland or intensively managed grasslands typically lose massive amounts of $N_r$ via gaseous emissions to the atmosphere or via nitrate leaching into soil and water bodies, (semi-) natural ecosystems such as forests and peatlands receive crucial $N_r$ input deposited from the atmosphere, thereby being

a controlling factor for their productivity, species composition, and their biosphere-atmosphere exchange of greenhouse gases (Fleischer et al., 2013; Hurkuck et al., 2014). Depending on regional emission sources and local surface characteristics, the ratio of dry to wet deposition may range from 1:2 to 2:1 (Simpson et al., 2006).

Micrometeorological flux measurements of non-reactive greenhouse gases using the eddy-covariance technique have become a common procedure for continuous observations of the biosphere-atmosphere exchange (Baldocchi, 2014). Hundreds of flux

towers are nowadays organized in regional or continental-scale networks such as ICOS (Integrated Carbon Observation System; Franz et al., 2018; Gielen et al., 2017) and follow standardized measurement protocols (Rebmann et al., 2018). Despite recent technological advances, particularly in the area of laser spectroscopy, flux measurements of $N_r$ compounds are still challenging and have largely remained experimental with limited campaign durations mainly due to high costs for devices, maintenance, and operation (Flechard et al., 2011). Other aspects challenging $N_r$ flux field campaigns are gas phase reactions

(Meixner, 1994) and gas-aerosol particle interactions (Wolff et al., 2010) within the chemical mixture of $N_r$, thereby requiring individual measurements of several parameters simultaneously. A comprehensive overview of commonly-used measurement techniques for a variety of $N_r$ compounds is given in the introductory section of Marx et al. (2012). A general issue with flux measurements using the eddy-covariance technique is a potential flux loss occurring in the high-frequency range because of different types of setups and response characteristics of instruments, e.g., closed *vs.* open-path instruments. These losses are

found to be significantly higher for $N_r$ than for carbon dioxide ($CO_2$) or water vapor flux measurements (*cf.* Fratini et al., 2012;



Ibrom et al., 2007; Mamadou et al., 2016; Moravek et al., 2019; Wintjen et al., 2020a) and can be up to 35% of an individually measured flux rate.

Models simulating plant-soil-atmosphere interactions of $N_r$ species are useful tools to investigate exchange patterns of different ecosystems. Land surface-atmosphere schemes (Massad et al., 2010; Wichink Kruit et al., 2010) can either be applied in local-scale studies (e.g., Schrader et al., 2016) or within the framework of chemical transport models such as DEPAC (DEPosition of Acidifying Compounds) (Erisman et al., 1994) within LOTOS-EUROS (LOng Term Ozone Simulation – EURopean Operational Smog) (van Zanten et al., 2010; Manders et al., 2017). Outputs of these models help understand ecosystem functioning and can further be used for gap filling in order to compile total nitrogen budgets, which form the basis of national inventories of air pollutants and their assessment reports. For a more robust model validation and to better facilitate investigations of the high degree of spatial and temporal variability of $N_r$ fluxes, a close coupling of modeling and measurement studies has been recently postulated (Schrader, 2019; Schrader et al., 2020). The latter studies also provide a state-of-the-art overview of current challenges and perspectives in $N_r$ modelling focusing on biosphere-atmosphere exchange of ammonia.

The aim of this paper is the release and description of campaign datasets of $N_r$ fluxes, which have been made publicly available in the Zenodo repository (http://doi.org/10.5281/zenodo.4513854). Four multi-months to multi-year field campaigns covering two different ecosystem types (peatland and forest) have been carried out within the research projects NITROSPHERE (Brümmer et al., 2019) and FORESTFLUX (Brümmer et al., 2020). We used a quantum cascade laser (QCL) absorption spectrometer and a custom-built converter called Total Reactive Atmospheric Nitrogen Converter (TRANC) connected upstream to a chemiluminescence detector (CLD) for measuring ammonia and total reactive nitrogen (hereafter named $\Sigma N_r$) fluxes, respectively. These datasets demonstrate the suitability of QCL and TRANC in eddy-covariance setups and – as in-situ high-resolution data of surface-atmosphere fluxes of $N_r$ compounds are still scarce – highlight potential applications such as model validation, understanding temporal dynamics of $N_r$ exchange, derivation of deposition velocities, and using model output as a gap-filling strategy for eddy-covariance datasets. In the here presented study, we follow the nomenclature of Marx et al. (2012), with the definition of $\Sigma N_r$ comprising all nitrogen-containing trace species, but excluding $N_2$ and $N_2O$ as these are non-reactive in the lower troposphere.

## 2 Field campaign sites

Datasets of reactive nitrogen ecosystem-atmosphere fluxes presented in this paper were acquired during four field campaigns between October 2012 and June 2018. Measurements took place at two different sites representing a natural forest (FOR) and a semi-natural peatland (WET). At each of the sites, both high-frequency QCL and TRANC measurements of $NH_3$ and $\Sigma N_r$, respectively, were conducted in eddy-covariance setups. These micrometeorological measurements were accompanied by monthly integrated low-cost denuder and filter samplers for a variety of $N_r$ compounds (Section 3.3). Further site information, campaign numbers, individual campaign lengths, and site acronyms used in this paper are given in Tables 1 to 3.



## 2.1 Bourtanger Moor site (WET)

The Bourtanger Moor is a natural park in the border region between Northwest Germany and Northeast Netherlands. The area was a wide-ranging and intact raised bog ecosystem in central Europe before being drained and cultivated in the seventeenth century (Casparie, 1993). Before rewetting was considered, the site had been moderately drained since the 1950s and constitutes nowadays one of the last  protected semi-natural peatlands in the region, whereas the surrounding raised bog complex of an approx. size of 700 ha is characterized by extensive peat excavation with drainage ditches and water table lowering primarily for the purpose of oil and gas drilling. Maximum accumulated peat depth is roughly 4 m. Main vegetation components are bog heather (*Erica tetralix*), purple moor grass (*Molinia caerulea*), cotton grass (*Eriophorum vaginatum* and *Eriophorum angustifolium*), coppices with some solitary birches (*Betula pubescens*) and Scots pines (*Pinus sylvestris*), and Sphagnum mosses. The Bourtanger Moor is surrounded by an area of intensive crop production and livestock breeding resulting in ~25 kg ha$^{-1}$ atmospheric nitrogen deposition annually. The ratio of dry to wet deposition has been found to be ~2:3 (Hurkuck et al., 2014) thereby substantially exceeding the critical N load of 5 to 10 kg ha$^{-1}$ yr$^{-1}$ prescribed for semi-natural peatlands. More site-specific information about the Bourtanger Moor can be found in the publications by Hurkuck et al. (2015; 2016) and references therein. Considerable seasonal variability in water table depth was observed with fully water-saturated soil in December and January and up to 60 cm below surface in late summer and early autumn averaging annually at around 10 cm. Annual average air temperature and precipitation for the period 1981 to 2010 at a nearby weather station from the German Meteorological Service in Lingen was 10.0°C and 799 mm, respectively.

## 2.2 Bavarian Forest site (FOR)

The Bavarian Forest site represents a remote location some tens of kilometres away from rather small anthropogenic emission sources providing the opportunity to study background concentration variability and natural flux exchange of a mixed-forest ecosystem. The site is located in the 'Forellenbach' catchment in the Bavarian Forest National Park, thereby being unmanaged and a tree composition of ~80% spruce (*Picea abies)* and ~20% beech (*Fagus sylvatica*) in the flux tower footprint. The forest was subjected to a massive bark beetle outbreak between 1995 and 2000 and is still recovering from the disturbance (Beudert et al., 2014). Stand height during the campaign was up to 20 m. Mean annual air temperature and precipitation according to records from the Bavarian Forest National Park Administration from the years 1978 to 2017 is 6.6°C and 1563 mm, respectively.

The Bavarian Forest National Park is part of several international observation networks. The most important are the Long-Term Ecological Research (LTER, https://data.lter-europe.net/, *last access: March 6, 2021*) network with the 'Forellenbach' catchment furthermore being part of the International Cooperative Program on Integrated Monitoring of Air Pollution Effects on Ecosystems (ICP IM) within the framework of the Geneva Convention on Long-Range Transboundary Air Pollution (https://unece.org/fileadmin//DAM/env/lrtap/WorkingGroups/wge/ im.htm, *last access: March 6, 2021*).



## 3 Measurement techniques

### 3.1 TRANC – Total Reactive Atmospheric Nitrogen Converter

The detection of concentrations of total reactive nitrogen ($\Sigma N_r$) were conducted with a manufactured in-house device named TRANC connected upstream to a chemiluminescence detector (CLD, model 780TR, ECO PHYSICS AG, Dürnten, Switzerland). The housing of the TRANC has an outer diameter of 12 cm and is 71 cm long. In all field campaigns the sonic anemometer was positioned upwind of the TRANC from the main wind direction, thereby keeping unintended disturbance of the natural micro-turbulence as low as possible. The measurement concept is based on two sequential conversion paths with

the first one being a thermal step and the second one being a catalytic step including the addition of carbon monoxide (CO) as a reducing agent. Reaction temperatures are 870 °C and 300 °C in an actively heated iron-nickel-chrome (FeNiCr) tube and a passively heated gold tube, respectively. Within the two paths, all reactive nitrogen compounds are converted to nitric oxide (NO), which is analysed in a downstream connected CLD at a frequency of 10 Hz (FOR site) or 20 Hz (WET site). Flow rate was kept constant at ~2.0 L min$^{-1}$ depending on site setup (*cf.* Table 2) by using a dry vacuum scroll pump (BOC Edwards

XDS10, Sussex, UK) connected to the CLD. Pressure reduction was assured through a critical orifice upstream of the CLD ensuring the required regime of approx. 14 mbar needed for the operation of the reaction chamber. The conversion efficiency of the TRANC under both laboratory conditions and in a long-term field experiment was tested by Marx et al. (2012). In the laboratory, the authors examined recovery rates for the gases $NO_2$ and $NH_3$, which were found to be 99 % and 95 %, respectively. A further test for a compound mixture of $NO_2$ and $NH_3$ during an eleven-month field campaign resulted in an average recovery

of 91 %. Detailed information on TRANC description and performance are presented in Ammann et al. (2012), Brümmer et al. (2013), Wintjen et al. (2020a), Zöll et al. (2019), and in Fig. 1 and 2.

### 3.2 QCL – Quantum Cascade Laser Spectrometer

Ammonia concentration measurements were conducted with a quantum cascade laser absorption spectrometer, model mini QC-TILDAS-76, Aerodyne Research, Inc., Billerica, MA, USA. The QCL was connected to a dry vacuum scroll pump

(TriScroll 600, Agilent Technologies, Santa Clara, USA) maintaining a flow rate through the system of approx. 17 L min$^{-1}$. Ammonia concentrations were measured in a 0.5 L multi-pass adsorption cell (76 m path length) at a frequency of 10 Hz. The signal was corrected for water vapor dilution using an online implementation, thereby providing mixing ratios of mol $NH_3$ per mol dry air. The QCL has a detection limit in the sub-ppb range (*cf.* McManus et al., 2008). Precision of the instrument is 0.042, 0.021, 0.016, and 0.010 ppb over 1, 10, 20, and 60 s, respectively. A specifically designed 'inertial inlet' box by the

manufacturer was installed upstream of the QCL using a 3.5 m heated (40 °C) anti-adhesive perfluoroalkoxy (PFA) tube to avoid unintended interactions of $NH_3$ with other gases, particles, and surfaces. More than 50% of the aerosols (>300 nm) were removed inside the inlet box through a split in the incoming air and a forced 180° turn of the air sample (Ellis et al., 2010; Ferrara et al., 2012; von Bobrutzki et al., 2010; Zöll et al., 2016). Additionally, an "active passivation" system injecting a surfactant vapor (perfluorooctylamine) to the inlet of the instrument to continuously coat the inlet walls was used for response





time improvements and minimizing wall losses (*cf.* Roscioli et al., 2016). A critical glass orifice maintained an operating pressure range of 4.6 to 6 kPa. Glass parts of the inlet box were cleaned prior to each campaign and at least once a month during field operation. The system was calibrated internally by the laser itself through alignment of the sampled $NH_3$ absorption peak to the standard of the HITRAN database (Rothman et al., 2009). Other than measurement height (*cf.* Table 2), the setup for ammonia concentration measurements at each campaign was identical to the one reported in Zöll et al. (2016), where also

details about instrument performance can be found.

### 3.3 Low-resolution samplers

#### 3.3.1 Passive samplers

During each of the campaigns, passive samplers of the IVL type (named after the Swedish Environmental Research Institute; Ferm, 1991) were used for monthly integrated $NH_3$ concentration measurements. IVL samplers had been satisfactorily tested

in past comparison studies (Dämmgen et al., 2010; Kirchner et al., 1999; Zimmerling et al., 2000). The sampler consists of a lens tube (length: 10 mm; opening: 20 mm; material: PP (Polypropylene)) that is attached vertically to the exposure. The upper tube opening is closed with a snap cover (material: PE (Polyethylene)) attached to a coated filter (company: Sartorius, Göttingen, Germany; material: cellulose; pore size: 0.45 μm). The lower opening is closed by a PTFE filter (company: Millipore, Darmstadt, Germany; diameter: 25 mm; pore size: 1.0 μm), which is stabilized on both sides with a stainless-steel

mesh (mesh size: 0.125 mm; wire diameter: 0.08 mm). PTFE membrane and steel mesh are pressed onto the tube using a second snap cover (material: PE), which has a punched opening of 20 mm and is the sample opening of the passive collector. For $NH_3$ separation, the cellulose filter was coated with citric acid. The mass of $NH_3$ being deposited on the coated filter depends on the molecular diffusion coefficient, exposure period, and design of the sampler (Dämmgen et al., 2010; Ferm, 1991). During laboratory analysis, impregnated filters were extracted with 5 ml ultrapure water using a shaking apparatus for

1 h before applying either segmented flow analysis or ion chromatography for $NH_3$ detection.

#### 3.3.2 DELTA denuder and filter samplers

Monthly integrated air concentrations of $NH_3$, $HONO$, $HNO_3$, particulate $NH_4$, and particulate $NO_3$ were measured with a custom-built DELTA (DEnuder for Long-Term Atmospheric sampling) system. A comprehensive description of the measurement principle is given in Sutton et al. (2001) and Tang et al. (2009). The system consists of four cylindrical denuders,

two of which are for the determination of acidic gases and two for ammonia. A filter pack captures nitrogen species in particulate matter. Air flow was recorded with a standard gas meter and was usually averaging 0.5 $m^3$ $d^{-1}$. In each of the campaigns, sampling height of the DELTA system was equal to the measurement height of the sonic anemometer (*cf.* Table 2). Basic denuders were coated with sodium carbonate and glycerol dissolved in water and methanol, whereas acidic denuders were coated with citric acid and glycerol and also being dissolved in water and methanol. The filter pack was a two-stage

holder using round cellulose filter (Whatman No. 1, diameter 25 mm), one coated with citric acid saturated in methanol, the



other one coated in potassium hydroxide solution in methanol. Sample processing and analysis were identical to the procedures used for KAPS denuder described in Dämmgen et al. (2010) and Hurkuck et al. (2014). Amount of substances on equally coated denuders were added.

### 3.4 Instrumentation for meteorological measurements

At all campaign sites, the same ultrasonic anemometer model (R3, Gill Instruments, Lymington, UK) was installed for the measurement of the 3-dimensional wind components ($u$, $v$, and $w$). From these signals wind speed, wind direction, friction velocity ($u_*$), and atmospheric stability were calculated. Measurement heights above ground were adapted to the respective vegetation type and are listed in Table 2. In every setup, both QCL inlet box and TRANC were mounted on a separate boom at the north side of the anemometer, which was in all cases an infrequent (<10%) wind direction. The position of the sample

air inlets was approx. 10 cm below and 30 cm north of the centre of the sonic anemometer array. Air temperature ($T_a$) and relative humidity (RH) were measured with a HC2S3 probe (Campbell Scientific, Logan, Utah, USA). Global radiation ($R_g$), i.e. total incoming shortwave radiation, was recorded by the upward facing pyranometer from a CNR4 net radiometer (Kipp&Zonen B.V., Delft, The Netherlands). An internal Pt-100 and a thermistor temperature sensor were used to correct for instrument heating. Precipitation was recorded with an automatic tipping-bucket rain gauge (Thies Clima, Göttingen,

Germany). During all campaigns, meteorological instruments were installed at the approx. same height as the sonic anemometer. Further details can be taken from the individual campaign publications listed in Table 2.

## 4 Data processing

### 4.1 Acquisition and flux calculation

The procedure of data collection was nearly the same at all sites. The software *EddyMeas*, a program of the software package

*EddySoft* (Kolle and Rebmann, 2007), was used for half-hourly recording of the measurements at all sites but with different time resolutions. In campaign 1 (WET-TRANC, Table 1), the sampling frequency was 20 Hz, whereas the data acquisition of sonic anemometer and respective gas analysers was set to a frequency of 10 Hz during campaigns 2 to 4 (WET-QCL, FOR-TRANC, FOR-QCL). Analog signals of the CLD, QCL, and anemometer were recorded at the sonic's interface and joined to a common data stream. Only at WET, $NH_3$ measurements were logged separately on the onboard computer of the QCL. Thus,

data from the QCL and the sonic had to be interpolated to a reference timestamp (*cf.* Zöll et al., 2016). Before calculating eddy-covariance fluxes (Aubinet et al. 2012; Burba, 2013), periods of maintenance, instrument malfunction, and obvious outliers were flagged based on visually checking instrumental performance and parameters such as flow rate, laser operating temperature, absorption cell pressure and temperature, and TRANC heating temperature. The software *EddyPro* (https://www.licor.com/env/support/EddyPro/topics/introduction.html) was used for eddy flux calculation. Raw data time

series were accounted for spikes after Vickers and Mahrt (1997), a 2-D rotation of the wind vector components was applied (Wilczak et al., 2001), and block averaging was performed. Half-hourly fluxes were calculated using the eddy-covariance



method. The time series of vertical wind speed and gas concentration are shifted against each other until the covariance is maximized. The number of samples a time series has to be shifted corresponds to the time lag. The physical time lag is the elapsed duration from a point in time the measurement of an air sample is done by the sonic anemometer until the gas

concentration of the same sample is recorded by the gas analyser after moving through the sample tubes. We used the option "*covariance maximization with default*" for determining time lags and their corresponding fluxes in the respective half hour. The default values, which are given in Table 2, as well as the range around the physical lag were based on theoretical considerations. Ranges were adapted to the highest time lag density and were mostly $\pm 3$ s around the default value. For quantitative investigation of the fluxes, several quality flagging criteria were applied. These were (i) removing flux and

concentration outliers, (ii) excluding values where $w$, $T$, or $\Sigma N_r$ concentration were larger than the mean plus $3 \times 1.96\sigma$ with $\sigma$ being the standard deviation of the variances, and (iii) discarding fluxes, which were attributed to very stable atmospheric conditions, i.e. $u_* < 0.1$ m s$^{-1}$ (*cf*. Zöll et al., 2019). Furthermore, (iv) only fluxes with flag "0" and flag "1" after Mauder and Foken (2006) were used. The contribution of flag "0" and flag "1" to available fluxes after application of visual screening of outliers (i), exclusions based on variances (ii) and low turbulence (iii) is given in Table 3. After application of all quality

criteria, 48% of the $\Sigma N_r$ fluxes were left for further flux analysis.

## 4.2 High-frequency damping corrections

High-frequency signals measured by any eddy-covariance setup are influenced by chemical, physical, or sampling processes leading to noise and to a reduced intensity in high or low-frequency parts of the signal. Thus, fluxes calculated from these filtered signals are attenuated by certain factor compared to signals, which would be measured by an ideal instrument,

especially in the high-frequency range. These losses are relatively large for closed-path instruments designed for measuring $N_r$ compounds, since these gases are highly reactive and some of them 'sticky' compared to inert gases like $CO_2$, $CH_4$, or $N_2O$. As chemical and physical characteristics of inert compounds are different from those of reactive gases, formerly developed high-frequency correction methods for inert gases are not necessarily suitable for $N_r$ compounds like $NH_3$ and $\Sigma N_r$. Wintjen et al. (2020a) proposed empirical methods for determining high-frequency damping factors of $\Sigma N_r$. Since site setups differed in

studies where flux measurements of various reactive compounds have been presented (e.g., Ammann et al., 2012; Brümmer et al., 2013; Ferrara et al., 2012; Moravek et al., 2019), several empirical methods had to be used for a precise quantification of the specific high-frequency damping. For the determination of damping factors of $\Sigma N_r$ for both WET and FOR site measurements, ICO, i.e. the in situ co-spectral method proposed by Wintjen et al. (2020a) was applied. $NH_3$ fluxes of WET site measurements were corrected by the ogive method following Ammann et al. (2006) (*cf*. Zöll et al., 2016). Median damping

factor ranges for the individual site setups are given in Table 2. A detailed description of the used damping correction method can be found in the cited publications.



### 4.3 Gap filling

Every set of eddy-covariance measurements usually reveals data gaps of various lengths after rigorous quality control, irrespective of the species of interest. These gaps may originate from situations where the basic assumptions of EC theory fail, e.g. under non-turbulent conditions or when fluxes represent an area (footprint) that is non-uniform or non-horizontal. Further violations arise when the average of fluctuations does not equal zero or when significant density fluctuations are observed (Aubinet et al., 2012). Other reasons for data gaps commonly include instrument failure or implausible spikes of unknown origin. With the here presented datasets we provide gap-filled time series of $NH_3$ and $\Sigma N_r$ to better facilitate estimations of the sites' nitrogen budgets.

#### 4.3.1 Data-driven gap-filling

A common approach for gap filling is the usage of statistical methods such as Look-Up Tables (LUT), Mean Diurnal Variation (MDV), or Non-Linear Regression (NLR) (Falge et al., 2001). These methods were originally developed for fluxes of carbon dioxide and/or water vapor and require short-term stability of exchange patterns for a sufficient quality of the gap-filled flux values. In case of $\Sigma N_r$ and $NH_3$, the exchange pattern can heavily vary over different time scales as these species usually exhibit lower autocorrelation and higher non-stationarity than for example carbon dioxide or water vapor. Thus, the application of data-driven methods to $\Sigma N_r$ fluxes is recommended for data gaps spanning only over a few days. LUT require that dependencies of the gas of interest on several parameters such as temperature, relative humidity, radiation, concentration, or turbulence-related parameters are well defined. Applying LUT on $\Sigma N_r$ becomes complicated since $\Sigma N_r$ describes the sum of several $N_r$ compounds. These gases and particles differ significantly in physical and chemical properties and in their contribution to $\Sigma N_r$. The ratio of the compounds varies throughout the annual cycle (Wintjen et al., 2020b) and further depends on site characteristics and surrounding conditions. Little is known about the overall dependency of $\Sigma N_r$ on the climatic parameters. We therefore decided to use MDV as gap-filling technique. In our dataset, the window for calculating the filled flux was set to 5 days before and 5 days after the gap in the original time series.

#### 4.3.2 Model-based gap-filling (FOR-$\Sigma N_r$ only)

Regular MDV gap-filling cannot account for long gaps due to a limited averaging window size. However, filling these gaps is a necessary step before calculating long-term budgets, which was one of the main goals of the measurement campaign at FOR. We therefore used a dry deposition inferential model as an additional gap-filling method during this campaign. The computer code DEPAC (DEposition of Acidifying Compounds; Van Zanten et al., 2010) version 3.21 was used in a stand-alone version, i.e., with input data measured at the site instead of modelled inside a chemistry transport model (CTM). We here used the variant of DEPAC that is used within the chemistry transport model LOTOS-EUROS (Manders et al., 2017) to fill gaps in the measured fluxes. It is slightly different from the one described in Van Zanten et al. (2010) in the sense that it respects co-deposition of $SO_2$ and $NH_3$ in the non-stomatal deposition pathway (Wichink Kruit et al., 2017), and uses a 1-month moving





average $NH_3$ concentration instead of instantaneous concentration measurements in the parameterisation of the stomatal compensation point (*cf.* Wichink Kruit et al., 2010). Aerodynamic and boundary resistances were modelled following Garland

(1977) and Jensen and Hummelshoj (1995, 1997), respectively, using stability corrections following Webb (1970) and Paulson (1970).

An approximation to the dry deposition of $\Sigma N_r$ that was used to fill gaps was calculated as the sum of the modelled dry deposition fluxes of $NH_3$, $NO$, $NO_2$, and $HNO_3$. Particle deposition was not modelled. Input data were available on a half-hour basis for most variables. These include temperature, relative humidity, global radiation, ambient pressure, and friction velocity,

as well as concentrations of the individual chemical compounds. Monthly DELTA-Denuder concentration measurements were used for $HNO_3$, and, during times of QCL malfunction, $NH_3$.

### 4.4 Dry deposition inferential modelling (WET-NH₃ only)

A dry deposition inferential model was used as an additional plausibility check in our first $NH_3$-QCL campaign at WET (Zöll et al., 2016). We used the parameterisation of a bi-directional canopy compensation point model of Massad et al. (2010),

following the works of Nemitz et al. (2001), in a single-layer big-leaf configuration (i.e., without explicitly modelling exchange with the ground-layer). The stomatal resistance was modelled using the approach of Wesely (1989), who implemented a dependency on temperature and radiation with a minimum stomatal resistance for $H_2O$ of 200 s m$^{-1}$; all other exchange resistances and parameters were set following the recommendations of Massad et al. (2010) with default parameters for semi-natural vegetation described therein. Exchange parameters at the leaf-layer (e.g., stomatal and non-stomatal resistance) were

calculated using temperature and relative humidity at the mean notional height of trace gas exchange (*cf.* Nemitz et al., 2009).

### 4.5 Uncertainty estimation

While understanding and reporting of uncertainty is not consistent across disciplines (Alekandrov, 2001), describing the quality of measurements is necessary to understand the level of confidence in any observation (Csavina et al., 2017). The eddy flux community usually distinguishes between random and systematic errors in tower-based measurements. Random errors may

arise from a variety of sources including (i) turbulence sampling errors, e.g., due to incomplete sampling of large eddies and the associated uncertainty in the calculated covariance between vertical wind velocity ($w$) and the scalar of interest ($c$), (ii) instrument errors in the measurement of $w$ and $c$, and (iii) uncertainty attributable to changes in wind direction and velocity which influence the footprint (Aubinet et al., 2012). Systematic errors originate from unmet assumptions and methodological challenges such as advection – preferentially over non-flat terrain – or errors resulting from instrument calibration and design

including high-frequency losses (Wintjen et al., 2020a) and additionally from those errors associated with data processing (e.g., through detrending, coordinate rotation, gap filling). While for some sources of uncertainty a reliable quantification is impossible without further observations, e.g., in the case of advective fluxes, other procedures mentioned above have been developed for $CO_2$ fluxes such as bias errors for annual sums of carbon budgets as a consequence of gap filling (Richardson



et al., 2008; Lucas-Moffat et al., 2018) and can therefore not be straightforwardly applied to reactive nitrogen compounds
presented in this study due to the different exchange behaviour of the gases.

For the here presented datasets we provide uncertainty estimates as follows:

- Is the originally measured half-hour flux value kept, i.e. has a quality flag of "0" or "1" been assigned based on the criteria of Mauder and Foken (2006), we calculated random uncertainty following Finkelstein and Sims (2001).
- When short gaps are filled with the MDV method, we use the standard error of the mean from the half hours in the
respective calculation window.
- In the FOR TRANC dataset, larger gaps were filled by DEPAC model estimates (Section 4.3.2). For those half hours, the uncertainty is given by

$$F_{unc,model} = \frac{\tilde{x}}{F_{model}} \tag{1}$$

with $\tilde{x}$ being the median of the distribution $F_{unc,meas01}/F_{meas01}$, which is based on the uncertainty of measured fluxes
with quality flags "0" or "1" ($F_{unc,meas01}$), i.e. the Finkelstein and Sims (2001) method as mentioned above, and originally measured fluxes where also quality flags "0" or "1" had been assigned ($F_{meas01}$).
- The uncertainty of cumulative fluxes ($F_{unccum}$; Fig. 6) was estimated following the commonly used method from the eddy flux community given in Pastorello et al. (2020), i.e. by

$$F_{unccum_i} = \sqrt{\sum_i^n (F_{unc,meas01_i})^2}. \tag{2}$$

**5 Data description**

The presentation and structure of the published campaign datasets were attempted to be as self-explanatory as possible. Tables 1 to 3 containing information on site description, technical details with data descriptors, and column explanations were published alongside the data time series in the Zenodo repository ([http://doi.org/10.5281/zenodo.4513854](http://doi.org/10.5281/zenodo.4513854)). In the following, we briefly highlight key data characteristics.

Campaign time series and mean diurnal courses of both concentrations and fluxes are shown in Fig. 3. The considerably different pollution climate between the peatland (WET), which is subjected to intensive agricultural land use in close vicinity to the measurement location (*cf.* Hurkuck et al., 2014), and the forest (FOR) within a protected national park, becomes obvious. While at WET mean campaign values of $\Sigma N_r$ and $NH_3$ were 21.7 and 15.1 ppb, respectively, forest campaign means were 5.2 ppb for $\Sigma N_r$ and 1.8 ppb for $NH_3$ (Fig. 4). At all campaigns, clear diurnal concentration patterns with unexpectedly low noise
were observed. Surprisingly, WET $NH_3$ and FOR $\Sigma N_r$ – two different sites and two different analysers – revealed an almost similar bimodal distribution with one peak around noon and an even larger peak in the late afternoon. Lowest numbers were found during the night. The only apparent difference between these campaigns were the mean daily amplitude. While concentrations at WET $NH_3$ stretched over ~7 ppb, the daily averaged span at FOR $\Sigma N_r$ was within 1 ppb. The diurnal





concentration pattern at FOR $NH_3$ was similar to that of FOR $\Sigma N_r$ and WET $NH_3$, but it was showing only one distinct peak
in the late afternoon. Different diurnal concentration dynamics were found at WET $\Sigma N_r$. While the averaged daily amplitude
was ~4 ppb, the peak was found in the early morning hours, whereas lowest values were observed shortly after midday.

During all campaigns both emission and deposition of nitrogen could be detected on a half-hourly time resolution (Fig. 3 and
5). Mean campaign fluxes at WET $\Sigma N_r$, WET $NH_3$, and FOR $\Sigma N_r$ were −5, −17, and −14 ng N $m^{-2}$ $s^{-1}$, respectively. Negative
numbers indicate an overall net deposition to the respective ecosystem. After rigorous QA/QC, we concluded that a reliable
calculation of $NH_3$ fluxes at FOR site could not be achieved. Due to the very low concentration level close to the detection
limit of the analyser (*cf*. Zöll et al., 2016), we observed insufficient variability in $NH_3$ concentrations leading to a low signal-
to-noise ratio making a covariance detection between vertical wind and $NH_3$ concentration impossible.

As for the concentration regime, we found clear diurnal flux patterns with relatively low noise in the signal, at least at WET
$NH_3$ and FOR $\Sigma N_r$. At WET $\Sigma N_r$, no average net nitrogen exchange was observed in the afternoon, while $\Sigma N_r$ uptake was
relatively stable around −7 ng N $m^{-2}$ $s^{-1}$ during the night. The flux pattern for $NH_3$ at WET was similar, but with near-neutral
exchange occurring a few hours earlier in the afternoon. Night-time $NH_3$ uptake was stronger around ~20 to 25 ng N $m^{-2}$ $s^{-1}$.
The diurnal flux pattern of $\Sigma N_r$ at FOR was approximately inverted to those at the WET site. Highest average uptake around
18 ng N $m^{-2}$ $s^{-1}$ was found between 9:00 and 15:00, whereas night-time uptake of $\Sigma N_r$ was considerably lower at around 10
ng N $m^{-2}$ $s^{-1}$.

The frequency distribution of concentrations and fluxes is shown in Fig. 5. While relatively narrow peaks with >50 % of the
overall data in only 2 to 3 concentration and flux bins were found at FOR campaigns, a wide distribution spanning over a large
range of bins was observed during the WET campaigns. At both sites $NH_3$, was peaking at the respective lower concentration
end, i.e. at 7 and 1 ppb for WET and FOR, respectively, whereas $\Sigma N_r$ was more broadly distributed towards higher
concentrations with less distinctive peaks. This could be expected as $NH_3$ is part of the $\Sigma N_r$ signal, thereby making the shape
of the frequency distribution a matter of the contribution strength of other $N_r$ compounds to the $\Sigma N_r$ signal. The different site-
specific shares are presented alongside the flux dataset in the form of monthly integrated HONO, $HNO_3$, aerosol $NH_4$, and
aerosol $NO_3$ measured by DELTA denuder and filter samplers. Further details for the WET and FOR sites can be found in the
publications of Hurkuck et al. (2014) and Wintjen et al. (2020b), respectively. In more than a third (38.3 %) of the campaign
time, net emission of $\Sigma N_r$ was observed at the peatland site. For $NH_3$, the distribution was shifted towards larger and more
frequent deposition (83.7 %). With 88.1 % of all fluxes being recorded as net reactive nitrogen uptake by the ecosystem, the
forest site revealed the most dominant deposition regime.

As campaign lengths differed substantially, total cumulative N exchange of the campaigns cannot be directly compared.
However, their sums and evolution offer a useful opportunity to assess atmospheric N loads over longer time spans (see Section
6). In the nine weeks of QCL measurements, 892 g N $ha^{-1}$ in the form of $NH_3$ were taken up by the peatland. At the same site,
net $\Sigma N_r$ uptake was even less (830 g N $ha^{-1}$) indicating both bidirectionality of $N_r$ compounds (e.g., NO is usually emitted,
$NO_2$ deposited, $NH_3$ can be emitted or deposited) and most likely a tendency towards N saturation at the site due to the historic



land use in the region over the past decades (*cf*. Hurkuck et al., 2014). The forest site was a clear sink for $\Sigma N_r$ with 10.6 kg N ha$^{-1}$ being recorded after 2.5 years of measurements.

## 6 Potential applications

Various useful applications arise from the datasets presented in this study. Information on nitrogen exchange has remained highly speculative or had to undergo comprehensive process-based or statistical modelling based on a number of vague assumptions in the past before a clear picture could be derived. Through technological advancements given and detailed in the introductory and methodological section above, the long-term continuous character in combination with a much higher temporal resolution, mainly led to novel opportunities understanding the functioning of ecosystems and their associated land

surface-atmosphere interactions. These are briefly outlined in the following paragraph.

*Improving process understanding through high temporal resolution*

- While information on sub-daily $N_r$ exchange had been challenging to get due to metrological limitations (*cf*. Sutton et al., 2009), half-hour fluxes from eddy-covariance measurements allow for investigating the mechanisms

governing the surface-atmosphere exchange on different temporal scales. For example, high temporal resolution helps to identify the bidirectional exchange of $NH_3$ (Nemitz et al., 2001; Zhu et al., 2015; Schulte et al., 2021) and reveals phases of net emission *vs*. net deposition (McGinn et al., 2016; Zöll et al, 2016; Wintjen et al, 2020b). A clearer picture can be derived about the interplay of stomatal (Geßler et al., 2002) and non-stomatal pathways (Schrader et al., 2016), the exchange between soil and the atmosphere, seasonal site-specific conditions, e.g.,

through vegetation senescence and decomposition of fallen leaves (Hansen et al., 2013) or periods of elevated evaporation from either water droplets on leaves (Hales and Drewes, 1979) or whole water bodies. Furthermore, although not presented in this study, short-term temporal dynamics of $N_r$ after field management like crop or grassland fertilization usually leading to immediate emission peaks (Martins et al., 2017) can be used for better quantification of nitrogen losses to the atmosphere (Brümmer et al., 2013; Ma et al., 2020).


*Integrated view of total $N_r$ dynamics through TRANC technology*

- While in the past, a laborious combination of several methods was required, TRANC data have the advantage that the sum of all relevant reactive nitrogen compounds is ready to use for further analysis by the measurement of a single signal. This provides a simple integrated overview representing the internally complex interactions of

$N_r$ species – such as NO emission from soil, atmospheric $NO_2$ or $HNO_3$ deposition, bidirectional $NH_3$ exchange or gas-particle interactions (Nemitz et al., 2004; Wolff et al., 2010), amongst others – all incorporated in one single number. Although TRANC data do not reveal single compound patterns, applying the TRANC technology



reduces effort and investments for studies focussing on $\Sigma N_r$ balances and characteristics (e.g., Zöll et al., 2019; Wintjen et al., 2020b).


*Assessment of Critical Level and Critical Load exceedance*

- Critical loads and levels are a tool for evaluating the risk of air pollution impacts to ecosystems and are widely used by environmental protection agencies (Wichink Kruit et al., 2014). With datasets presented in this study, a robust assessment becomes possible through information on both concentration and flux averages as well as their total deposition sums over longer time periods. For example, ombrotrophic bogs like the Bourtanger Moor (WET site) represent ecosystems that are most sensitive to increased atmospheric nitrogen input (Bobbink et al., 2010). An exceedance of a certain deposition threshold (e.g., 5 kg N ha$^{-1}$ yr$^{-1}$ for bogs) may result in an invasion by more nitrophilous grasses and trees (e.g. *Molinia caerulea*, *Betula pubescens*) and a decline in the native ecosystem-specific species (Tomassen et al., 2003; Hurkuck et al., 2014). Fig. 4 and 6 highlight the potential of campaign data being used in local validations of possible critical level and critical load exceedances. In our cases, we found that ambient concentrations at a peatland site are similar to those at highly fertilized arable land (*cf.* Brümmer et al., 2013). Furthermore, unmanaged remote forest sites (like FOR) some tens of kilometres away from emission sources offer an opportunity to investigate the natural exchange mechanisms of trees with the atmosphere.


*Determination of controlling factors*

- A further potential application is the analysis of controlling factors of reactive nitrogen exchange. Fluxes have been shown to be highly non-linear (e.g. Flechard et al., 2013) in space and time, which complicates the development of surface-atmosphere exchange schemes. On the basis of the FOR dataset, Wintjen et al. (2020b) demonstrate that mean diurnal flux courses stratified by concentration, temperature, moisture or surface wetness reveal the strongest dependence on the actual $\Sigma N_r$ concentration regardless of flux direction, i.e. situations of deposition or emission. Deposition velocity ($v_d$), however, was invariant with changing concentration, temperature, and relative humidity under dry leaf conditions, whereas wet leaf surfaces led to increased atmospheric resistances ($R_a$) and decreased $v_d$, thereby reducing the $\Sigma N_r$ flux magnitudes.


*Verification of annual $N_r$ budgets and management strategies*

- Long-term data on average exchange fluxes help establish nitrogen budgets and provide an essential piece in the nitrogen cycle, particularly with regard to its temporal resolution. Cumulating all half-hourly fluxes like in Fig. 6 showcase a robust way to quantify total nitrogen input or loss from an ecosystem over a certain time span. A number of applications can be drawn from this information, e.g. (cross-)validation of models or verification of



annual nitrogen budgets. Furthermore, management practices like proper timing of fertilization events with their respective amounts of nitrogen content can be derived. In Brümmer et al. (2013) the authors conclude that the last of three fertilizer inputs at a Thuringian crop site was likely to be unnecessary as much of the added nitrogen was not properly taken up by already senescing wheat plants and was immediately lost to the atmospheric, thereby
causing an additional yet avoidable burden to local air quality.

*Investigation of $N_r$ deposition effects on greenhouse gas exchange and carbon storage capacity of ecosystems*

- Finally, these datasets can be used to analyse interactive effects with other components of biogeochemical cycles. Particular interest is placed on how nitrogen deposition may alter the carbon sink strength over different land-
use types. While ecosystems such as forests may benefit from an increased nitrogen input from the atmosphere (de Vries et al., 2009; Flechard et al., 2020; Fleischer et al., 2013; Bala et al., 2013), others like nitrogen-limited peatlands were shown to respond with inhibited carbon uptake as a consequence of surplus available nitrogen (Limpens et al., 2011). The here presented datasets may offer the opportunity to validate those results and help find potential – yet unknown – effects on shorter time scales.

**7 Conclusions and outlook**

Our datasets demonstrate the suitability of QCL (Quantum Cascade Laser) and TRANC (Total Reactive Atmospheric Nitrogen Converter) to measure eddy-covariance fluxes of $NH_3$ and $\Sigma N_r$, respectively. In terms of stability, practicality and ease of operation, the standardization of field setups and data post-processing of reactive nitrogen measurements still have a highly experimental character, thereby being two decades behind those of inert greenhouse gas measurements, which are nowadays
organized in continental-scale flux networks like for example ICOS in Europe (Heiskanen et al., 2021) or the National Ecological Observation Network (NEON, Metzger et al., 2019) in North America. Rigorous estimation of high-frequency flux losses and their respective application are inevitable. Dry deposition model schemes can be used for measurement data gap filling, while measured data can be used for model validation and derivation of deposition velocities. As underlying mechanisms of biosphere-atmosphere exchange of $NH_3$ and $\Sigma N_r$ are not yet fully understood, continuous long-term
measurements are beneficial for understanding temporal dynamics and their controls. A number of potential applications arise from the datasets presented in this study such as the assessment of critical loads, the verification of management strategies or a proper analysis of $N_r$ deposition effects on greenhouse gas exchange and carbon storage capacity of ecosystems. We recommend future QCL and TRANC measurement campaigns at fertilized arable sites for the provision of verification opportunities for emission factors used in national emission inventories. Furthermore, selected sites in long-term research
infrastructure networks should be equipped with (low-cost) measurement devices for reactive nitrogen compounds as it has been recently shown by Schrader et al. (2020) that a link between stomatal conductance derived from $CO_2$ measurements may

be useful for estimating ammonia dry deposition, which would in turn offer a potential validation tool for regional to national-scale deposition maps.

## 8 Data availability and structure

The presented datasets have been placed into the Zenodo data repository and are accessible under http://doi.org/10.5281/zenodo.4513854. The structure of the dataset is given in Table 3 of this manuscript. Beside auxiliary meteorological data like air temperature, relative humidity, global radiation, wind speed, wind direction, friction velocity, atmospheric stability, and precipitation, we make field campaign measurements and model output of concentrations and fluxes of total reactive nitrogen and ammonia on half-hourly basis publicly available.

**Author contributions**

CB, JJR, JPD, PW, and FS designed the study and performed QA/QC on datasets during campaigns and post-processing. JJR and JPD designed the field setups and developed data acquisition procedures. JJR, JPD, CB, and PW conducted the measurements and BB provided scientific advice and logistical support for the Bavarian forest campaign. MS and FS developed and provided model schemes used for flux data analyses and gap filling. CA and PW developed and implemented correction

procedures for high-frequency flux losses and helped optimizing analyser performance. FS, PW, JJR, and CB established the database. CB, PW, and FS wrote the manuscript and all authors provided substantial input.

**Competing interests**

The authors declare that they have no conflict of interest.

**Acknowledgements**

The authors acknowledge funding through the German Federal Ministry of Education and Research (BMBF) within the framework of the Junior Research Group NITROSPHERE under support code FKZ 01LN1308A and through the German Environmental Protection Agency (UBA) for the FORESTFLUX project under grant number FKZ 3715512110. Further support was provided by the German Federal Ministry of Food and Agriculture (BMEL) through the Thünen Institute of Climate-Smart Agriculture. We highly appreciate data QA/QC, data processing, and field campaign support by Undine Zöll.

Ute Tambor, Andrea Niemeyer, and Dr. Daniel Ziehe are thanked for laboratory analyses of denuder and filter samplers. David D. Nelson and Mark Zahniser introduced the authors to the operation of the laser spectrometer and supported remote maintenance procedures for the instrument, which is highly appreciated.



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





**Tables**

**Table 1: Overview of field campaigns with $\Sigma N_r$ referring to the sum of all reactive nitrogen compounds, TRANC being the Total Reactive Atmospheric Nitrogen Converter, QCL being a quantum-cascade laser spectrometer, and CLD being a chemiluminescence detector.**

**Table 2: Setups and data characteristics with $\Sigma N_r$ referring to the sum of all reactive nitrogen compounds, TRANC being the Total Reactive Atmospheric Nitrogen Converter, QCL being a quantum-cascade laser spectrometer, and CLD being a chemiluminescence detector.**

**Table 3: Dataset structure and description.**

**Figures**

**Figure 1: Map of campaign locations and setup pictures.**

**Figure 2: Exemplary sketch of instrumentation with QCL and TRANC being simultaneously operated, not true to scale; QCL is quantum cascade laser, CLD is chemiluminescence detector, and TRANC is total reactive atmospheric nitrogen converter.**

**Figure 3: Campaign time series of half-hourly (coloured) and daily mean (white squares with black edges/grey squares) concentrations (first column) and fluxes (third column); second and fourth column represent mean diurnal courses of concentrations and fluxes of the specific entire campaign period, respectively; black lines are 3-hr moving averages; positive flux values indicate emission, negative flux values indicate deposition of nitrogen; error bars are the standard error of the mean.**

**Figure 4: Summary of campaign concentrations and fluxes. White circles with black edges represent the campaign mean, horizontal lines within boxes indicate the median, vertical box dimensions indicate lower and upper quartile values, whiskers represent the interquartile range and outliers from this range are plotted as grey crosses; positive flux values indicate emission, negative flux values indicate deposition of nitrogen.**

**Figure 5: Frequency distribution of half-hourly campaign concentrations and fluxes; n=9005 at WET $\Sigma N_r$ campaign, n=2916 at WET NH₃ campaign, n=42635 at FOR $\Sigma N_r$ campaign, n=23046 at FOR NH₃ campaign; positive flux values indicate emission, negative flux values indicate deposition of nitrogen.**

**Figure 6: Cumulative campaign fluxes; larger gaps at the WET site were not further interpolated; negative cumulative flux values indicate overall deposition of nitrogen; grey shaded areas represent uncertainty (see Section 4.5 for details).**





**Table 1: Overview of field campaigns with ΣN$_r$ referring to the sum of all reactive nitrogen compounds, TRANC being the Total**
890 **Reactive Atmospheric Nitrogen Converter, QCL being a quantum-cascade laser spectrometer, and CLD being a chemiluminescence detector.**

| Campaign number | Site name | Site acronym | Coordinates | Altitude | Land use | Site description | Campaign period | Type of measurement |
|---|---|---|---|---|---|---|---|---|
| 1 | Bourtanger Moor | WET | 52°39'N, 7°11'E | 14 m a.s.l. | Peatland | Peat bog in natural park | 10.2012 to 08.2013 | ΣN$_r$ with TRANC-CLD |
| 2 | Bourtanger Moor | WET | 52°39'N, 7°11'E | 14 m a.s.l. | Peatland | Peat bog in natural park | 02.2014 to 05.2014 | NH$_3$ with QCL |
| 3 | Bavarian Forest | FOR | 48°56'N, 13°25'E | 807 m a.s.l. | Forest | Bavarian Forest National Park with 80% spruce and 20% beech | 10.2015 to 06.2018 | ΣN$_r$ with TRANC-CLD |
| 4 | Bavarian Forest | FOR | 48°56'N, 13°25'E | 807 m a.s.l. | Forest | Bavarian Forest National Park with 80% spruce and 20% beech | 10.2015 to 06.2018 | NH$_3$ with QCL |





**Table 2: Setups and data characteristics with $\Sigma N_r$ referring to the sum of all reactive nitrogen compounds, TRANC being the Total Reactive Atmospheric Nitrogen Converter, QCL being a quantum-cascade laser spectrometer, and CLD being a chemiluminescence detector.**

| Campaign | 1 | 2 | 3 | 4 |
|---|---|---|---|---|
| **Site** | WET | WET | FOR | FOR |
| **Species investigated** | $\Sigma N_r$ | $NH_3$ | $\Sigma N_r$ | $NH_3$ |
| **Analytical device** | TRANC with CLD | QCL | TRANC with CLD | QCL |
| **Analyzer model** | CLD 780 TR, ECO PHYSICS AG, Dürnten, Switzerland | mini QC-TILDAS-76, Aerodyne Research, Inc., Billerica, MA, USA | CLD 780 TR, ECO PHYSICS AG, Dürnten, Switzerland | mini QC-TILDAS-76, Aerodyne Research, Inc., Billerica, MA, USA |
| **Sonic anemometer model** | GILL-R3, Gill Instruments, Lymington, UK | GILL-R3, Gill Instruments, Lymington, UK | GILL-R3, Gill Instruments, Lymington, UK | GILL-R3, Gill Instruments, Lymington, UK |
| **Measurement height** | 2.5 m | 2.5 m | 30.0 m | 30.0 m |
| **Tube length** | 12.0 m | 3.0 m | 45.0 m | 3.0 m |
| **Flow rate** | 2.0 L min$^{-1}$ | 17.0 L min$^{-1}$ | 2.1 L min$^{-1}$ | 17.0 L min$^{-1}$ |
| **Mean lag time** | 2.5 s | 0.79 s | 20.0 s | - |
| **Damping factor range** | 0.74-0.45 | 0.67 | 0.90-0.62 | - |
| **Percentage of flag 0 and 1 data** | 89.7 | 84.4 | 84.4 | - |
| **Percentage deposition fluxes** | 61.7 | 83.7 | 88.1 | - |
| **Percentage emission fluxes** | 38.3 | 16.3 | 11.9 | - |
| **Reference** | Wintjen et al. (2020) | Zöll et al. (2016) | Zöll et al. (2019); Wintjen et al. (2020a,b) | - |



**Table 3: Dataset structure and description.**

| Column header name | Unit/Format | Description |
|---|---|---|
| Date_Time_WET_TRANC | (yyyy-mm-dd HH:MM) | Time stamp of specific campaign, for acronyms see Table 1 |
| Ta | (degC) | Air temperature |
| RH | (%) | Relative humidity |
| Rg | (W m$^{-2}$) | Global radiation |
| WS | (m s$^{-1}$) | Wind speed |
| WD | (deg) | Wind direction |
| ustar | (m s$^{-1}$) | Friction velocity |
| zeta | (-) | Atmospheric stability, $(z\text{-}d)/L$ |
| precipitation | (mm) | Precipitation (rain + snow) |
| total_Nr_conc | (ppb) | Concentration of total reactive nitrogen ($\Sigma N_r$) |
| total_Nr_flux_meas_q0 | (ng m$^{-2}$ s$^{-1}$) | Flux of total reactive nitrogen ($\Sigma N_r$), only measured fluxes with quality flag 0 |
| total_Nr_flux_meas_q01 | (ng m$^{-2}$ s$^{-1}$) | Flux of total reactive nitrogen ($\Sigma N_r$), only measured fluxes with quality flag 0 and 1 |
| total_Nr_flux_meas_q012 | (ng m$^{-2}$ s$^{-1}$) | Flux of total reactive nitrogen ($\Sigma N_r$), only measured fluxes with quality flag 0 and 1 and 2 |
| total_Nr_flux_gf_MDV | (ng m$^{-2}$ s$^{-1}$) | Flux of total reactive nitrogen ($\Sigma N_r$), gap-filled with MDV method, see Section 4.3.1 |
| total_Nr_flux_gf_model | (ng m$^{-2}$ s$^{-1}$) | Flux of total reactive nitrogen ($\Sigma N_r$), gap-filled with model, see Section 4.3.2 |

900



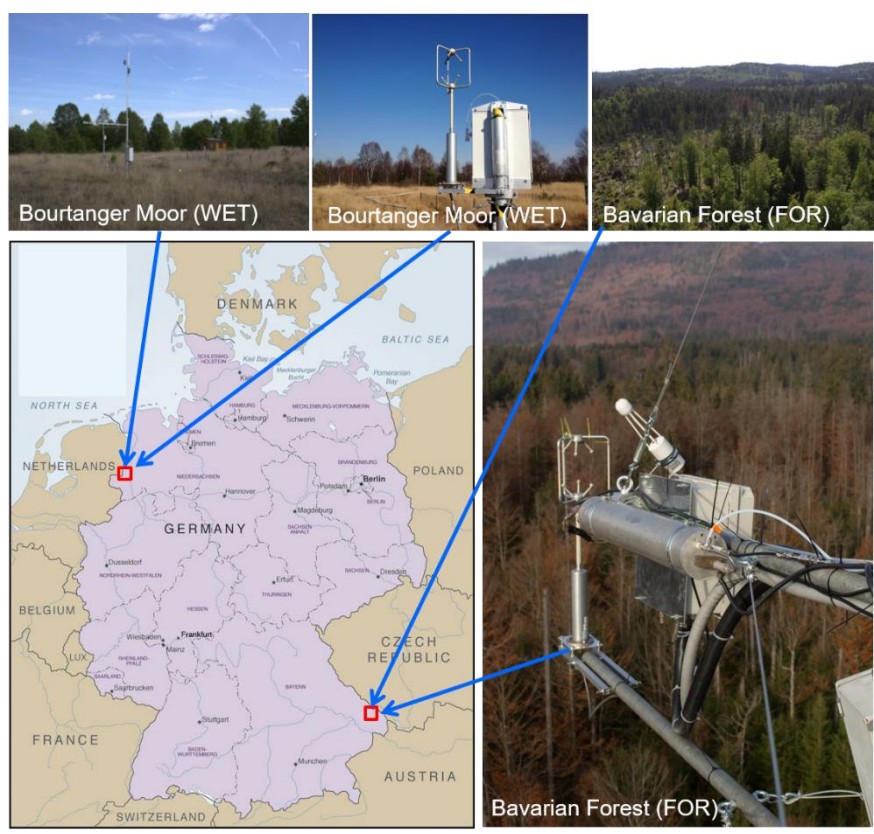

http://www.smartraveller.gov.au/zwiki/images/regions/maps/jpeg
/Germany.jpg

905 **Figure 1: Map of campaign locations and setup pictures.**



**Figure 2: Exemplary sketch of instrumentation with QCL and TRANC being simultaneously operated, not true to scale; QCL is quantum cascade laser, CLD is chemiluminescence detector, and TRANC is total reactive atmospheric nitrogen converter.**

**Figure 3: Campaign time series of half-hourly (coloured) and daily mean (white squares with black edges/grey squares) concentrations (first column) and fluxes (third column); second and fourth column represent mean diurnal courses of concentrations and fluxes of the specific entire campaign period, respectively; black lines are 3-hr moving averages; positive flux values indicate emission, negative flux values indicate deposition of nitrogen; error bars are the standard error of the mean.**

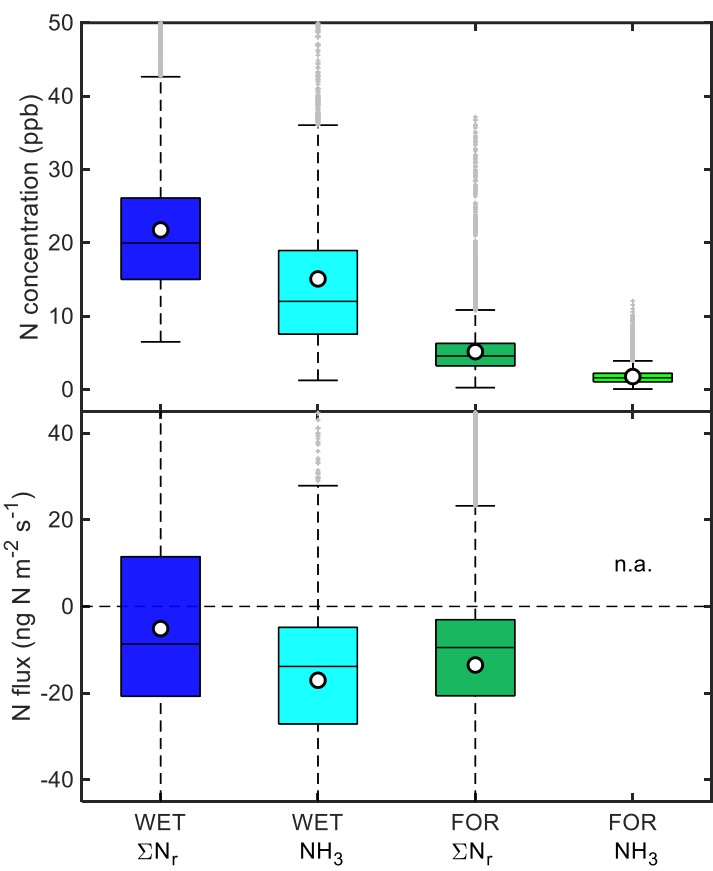

**Figure 4: Summary of campaign concentrations and fluxes. White circles with black edges represent the campaign mean, horizontal lines within boxes indicate the median, vertical box dimensions indicate lower and upper quartile values, whiskers represent the interquartile range and outliers from this range are plotted as grey crosses; positive flux values indicate emission, negative flux values indicate deposition of nitrogen.**



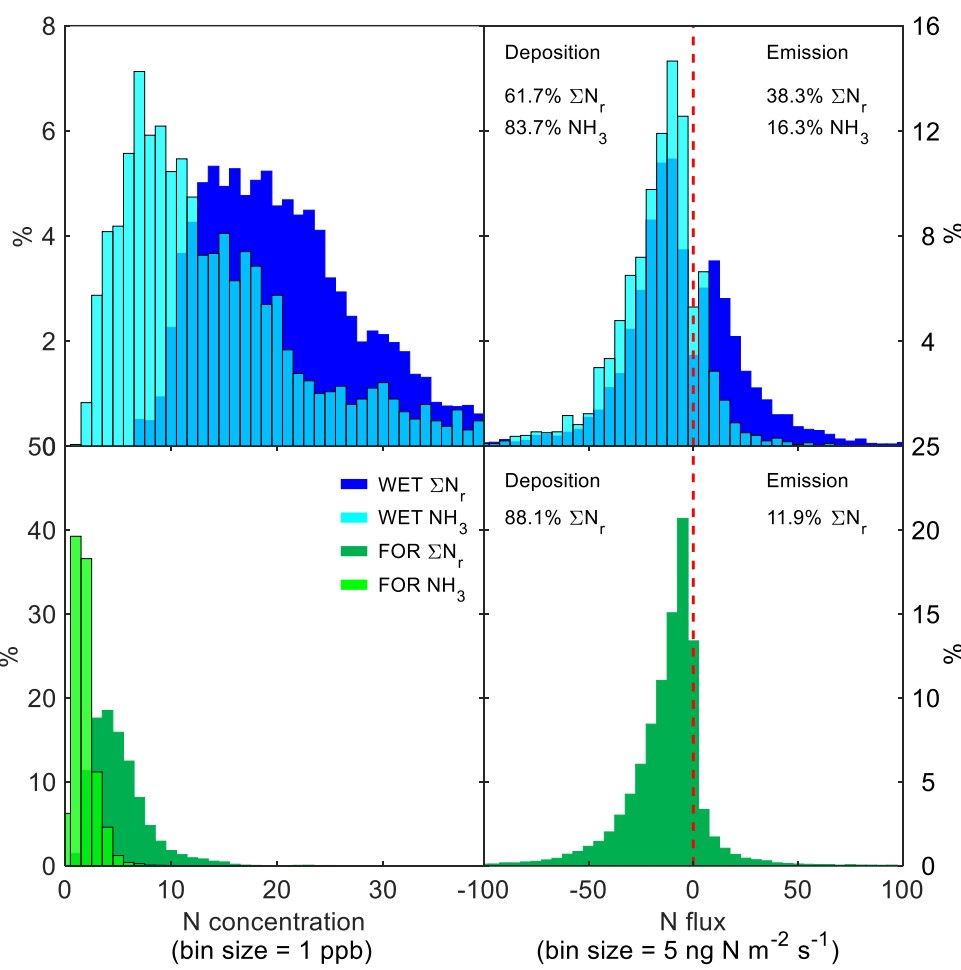

**Figure 5: Frequency distribution of half-hourly campaign concentrations and fluxes; n=9005 at WET $\Sigma N_r$ campaign, n=2916 at WET NH₃ campaign, n=42635 at FOR $\Sigma N_r$ campaign, n=23046 at FOR NH₃ campaign; positive flux values indicate emission, negative flux values indicate deposition of nitrogen.**

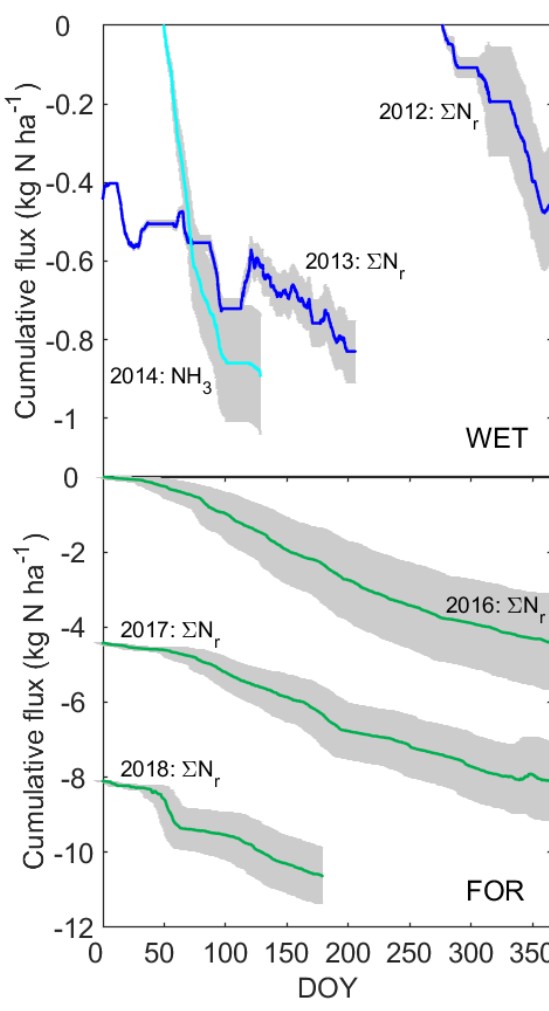

935

**Figure 6: Cumulative campaign fluxes; larger gaps at the WET site were not further interpolated; negative cumulative flux values indicate overall deposition of nitrogen; grey shaded areas represent uncertainty (see Section 4.5 for details).**