# Peer review of "Reactive nitrogen fluxes over peatland and forest ecosystems using micrometeorological measurement techniques"

_Earth System Science Data, 2021_

## Author Comment (AC1)

**Response to Referee Comments**

Manuscript "Reactive nitrogen fluxes over peatland and forest ecosystems using micrometeorological measurement techniques"

essd-2021-85

The authors sincerely thank the two reviewers for their thorough review. This document lists our responses to their comments.

Comment on essd-2021-85
Anonymous Referee #1

Referee comment on "Reactive nitrogen fluxes over peatland and forest ecosystems using micrometeorological measurement techniques" by Christian Brümmer et al., Earth Syst. Sci. Data Discuss., https://doi.org/10.5194/essd-2021-85-RC1, 2021

**Reviewer comment 1.1:**

The auhors present the almost annual high-resolution (half-hourly) total Nr and NH3 flux datasets at two representive ecosystems. The dataset may be useful for model validation for atmospheric chemistry and land surface processes. However, considering publication I have to ask the authors to answer the following questions and comments;

- Could you demonstrate the novelty of the datasets more strongly, via comparing with the prior studies in terms of temporal resolution, data length, data quality, site characteristics (vegetation type)? For example, what about the quality or uncertainty of other past eddy covariance study for Nr over the vegetation compared with your datasets? How long is the longest record of past datasets, and where and what vegetation type?

**Authors' response 1.1:**

We appreciate the comment and agree that highlighting the novelty of the here presented datasets should be mentioned more strongly. The main reasons why there are not many eddy-covariance (EC) campaigns of reactive nitrogen (Nr) compounds and why campaign lengths are generally limited were already given in lines 51-62. These are – amongst others – high costs for maintenance, operation, and instruments, gas phase reactions, gas-aerosol interactions, basically a need for individual measurements for several compounds simultaneously as well as damping issues due to high-frequency losses.

We will further add an overview of reported EC studies of  $N_r$  compounds from literature, see Table R1 at the end of this document.

It can be derived that

- Most of the other presented campaigns are very limited in lengths. The only exceptions are the Munger et al. (1996) and the Horii et al. (2006) papers that explicitly mention the determination of a flux budget as an aim of their study.
- Most of the other studies follow different aims, some intend to test the suitability of the presented analyzer for EC measurements, others focus on correction factors or error analysis, while some are investigating processes of tropospheric chemistry.
- Many studies do not explicitly give flux uncertainties or detection limits for their systems. In those studies where numbers are mentioned, a large range of values is reported.

- No other EC  $\Sigma N_r$  (i.e. total  $N_r)$  studies were found except for those by authors from this study.

We suggest to put the table into the Supplement of the article as some readers may find the information useful.

Additional text will be added in line 82 to properly link the novelty of the here presented datasets with the literature overview:

"The reader is referred to Table S1 in the supplementary section where an overview of previous eddy-covariance campaigns of different  $N_r$  compounds is given. The table highlights the limitations of campaign lengths, a wide range of flux uncertainties, and mainly different aims of previous studies such as testing the suitability of a certain analyzer, investigating processes of tropospheric chemistry, or derivation of correction factors."

**Reviewer comment 1.2:**

Another example is green-house gas flux; other ESSD papers demonstrated typically for greenhouse gas flux for longer time period (> several ten years), which enables us to discuss the annual carbon budget. Actually you mentioned about the impact of nitrogen deposition on carbon storage in 1443-448 in p15, but your WET site dataset for instance is collected for less than 1 yr and could not used for annual deposition/emission. Although this is partially described in Introduction section, more explanation is still required.

**Authors' response 1.2:**

It is true that we stress the possibility of the presented datasets for investigating potential effects of N deposition on CO2 exchange. We agree that even for the >2.5-yr period at the forest site, it is hardly possible to indicate trends in the respective annual balances, let alone some sort of robust analysis on this time scale would be meaningful. It was meant that continuous Nr flux datasets like the ones that are here presented, offer a useful tool to investigate the relationship between N deposition and CO2 exchange when carried out over multiple years. To clarify this, we rephrase line 443f. accordingly:

"Finally, continuous  $N_r$  flux datasets like the ones presented in this study can be used to analyse interactive effects with other components of biogeochemical cycles when carried out over multiple years."

Regarding the length of our presented datasets in comparison to typical greenhouse gas flux times series, it becomes clear that with the above-mentioned reasons such as the fact that campaigns have largely remained experimental due to high costs for devices, maintenance, and operation or the gas phase reactions requiring individual measurements of several species simultaneously (authors' response 1.1, also lines 51-62), the community is just simply not there yet where the  $CO_2$  flux community is. It is the aim of this paper to provide a first piece into a direction of more and longer Nr flux measurements and should motivate other groups to work on joint initiatives as well as consolidated longer-term monitoring programs of Nr.

**Reviewer comment 1.3:**

- Please add the following data to your repository or if available from other projects (e.g., LTER, fluxnet, ...) please add these link address to your repository website: net radiation, sensible and latent heat, CO2 fluxes over the canopy, soil temp and moisture (for model validation), atmospheric pressure, precipitation, downward long-wave radiation (for model input), plant functional type (PFT), LAI, canopy height, leaf trait, soil type and texture (for input parameters). Management (fertilizer practice) data, too. They are

generally required for land surface modeling in addition to your concentration and flux data.

**Authors' response 1.3:**

We agree that providing the requested data is useful for running land-surface models. We checked the availability and will add those datasets to the repository that we either measured ourselves or where we have access to, see Table R2 for details.

Table R2: Availability of additional datasets to be added to the repository website from the peatland (WET) and the forest (FOR) site, where "x" indicates available and "o" indicates that these data will not be added as they were either not recorded or have an insufficient quality due to bad instrument performance. Fertilization is of no relevance at both sites (unmanaged).

|                  | WET        | FOR                |  |
|------------------|------------|--------------------|--|
| Rn               | x          | 0                  |  |
| LE               | x          | 0                  |  |
| Н                | x          | x                  |  |
| F CO2 | x          | (x), only 3 months |  |
| Ts               | x          | 0                  |  |
| SWC              | o, but WTD | 0                  |  |
| P bar | x          | x                  |  |
| Precipitation    | x          | x                  |  |
| Rlong,down       | x          | 0                  |  |
| PFT              | х          | х                  |  |
| LAI              | 0          | (x), modeled       |  |
| Canopy height    | х          | х                  |  |
| Soil type        | x          | X                  |  |
| Soil texture     | х          | x                  |  |
| Fertilization    | n.a.       | n.a.               |  |

**Reviewer comment 1.4:**

- I think your low-resolution data can be used for validation of your high-resolution data over time series and/or scatter plots. Currently this seems to be used just for dry deposition estimation though.

**Authors' response 1.4:**

We thank the reviewer for the suggestion and support the idea of verifying the agreement between monthly integrated DELTA (or passive sampler, PS) concentrations and averaged QCL values for the corresponding exposition periods. However, we think that the high-resolution data should be the reference and not the other way around due to precision, accuracy and rigorous calibration of the QCL and no further error sources from post-field wet-chemical lab analyses involved in the DELTA and passive sampler procedures.

For fluxes, a validation of N deposition estimation with a model using monthly concentration input data is not recommended as the time resolution does not resolve micrometeorological variability such as diurnal courses of temperature, radiation, friction velocity or humidity. This is extensively shown and discussed in the publication of Schrader et al. (2018) and would not add any new insight to the here presented dataset publication.

We limit our analysis to the NH3 campaigns at the WET and the FOR site, because for  $\Sigma N_r$  at the WET site, DELTA and PS exposition did not overlap with the campaign period of the TRANC and at the FOR site the same topic is discussed in detail in the publications by Wintjen et al. (2022a, 2022b) for  $\Sigma N_r$ .

Figures R1 to R5 show the respective comparison of QCL vs. low resolution data from DELTA or PS. Data from the latter two low-resolution methods were consistently underestimated compared to QCL values. It must be noted that at the WET site, the data gap in QCL measurements in April 2014 led to a larger difference between QCL and DELTA as the values were generally decreasing from the early April peak due to fertilization on adjacent agricultural sites, hence biasing the QCL average. A full time series would have probably led to a slope much closer to 1 in Figure R2 as was found in the comparison between QCL and DELTA at the FOR site (*cf.* Figure R4).

We suggest adding these figures to the supplementary section and will extend lines 328f. in Section 5 (Data description) as follows:

"In the following, we briefly highlight key characteristics of the high-resolution TRANC and QCL data. A comparison with low-resolution concentration data from DELTA denuders and passive samplers is given in the supplementary section."

Figure R1: Concentration time series of  $NH_3$  at the peatland (WET) site. Horizontal red lines correspond to the exposition time of the DELTA denuders. For better comparability, averages of the QCL are shown in blue for the same periods.

Figure R2: Scatter plot of NH3 concentration from QCL and DELTA denuders corresponding to identical periods at the peatland (WET) site.

---

## Author Response (AR1)

**Point-by-Point Response to all Referee Comments**

Manuscript "Reactive nitrogen fluxes over peatland and forest ecosystems using micrometeorological measurement techniques"

essd-2021-85

The authors sincerely thank the two reviewers for their thorough review. This document provides a point-by-point response to all referee comments. Changes made in the manuscript are given in blue font. Line numbers correspond to the version where changes are tracked.

Comment on essd-2021-85
**Anonymous Referee #1**

Referee comment on "Reactive nitrogen fluxes over peatland and forest ecosystems using micrometeorological measurement techniques" by Christian Brümmer et al., Earth Syst. Sci. Data Discuss., https://doi.org/10.5194/essd-2021-85-RC1, 2021

**Reviewer comment 1.1:**
*The auhors present the almost annual high-resolution (half-hourly) total Nr and NH3 flux datasets at two representative ecosystems. The dataset may be useful for model validation for atmospheric chemistry and land surface processes. However, considering publication I have to ask the authors to answer the following questions and comments;*

*- Could you demonstrate the novelty of the datasets more strongly, via comparing with the prior studies in terms of temporal resolution, data length, data quality, site characteristics (vegetation type)? For example, what about the quality or uncertainty of other past eddy covariance study for Nr over the vegetation compared with your datasets? How long is the longest record of past datasets, and where and what vegetation type?*

**Authors' response 1.1:**
We appreciate the comment and agree that highlighting the novelty of the here presented datasets should be mentioned more strongly. The main reasons why there are not many eddy-covariance (EC) campaigns of reactive nitrogen ($N_r$) compounds and why campaign lengths are generally limited were already given in lines 51-62. These are – amongst others – high costs for maintenance, operation, and instruments, gas phase reactions, gas-aerosol interactions, basically a need for individual measurements for several compounds simultaneously as well as damping issues due to high-frequency losses.
We have further compiled an overview of reported EC studies of $N_r$ compounds from literature, see Table R1 at the end of this document.

It can be derived that
- Most of the other presented campaigns are very limited in lengths. The only exceptions are the Munger et al. (1996) and the Horii et al. (2006) papers that explicitly mention the determination of a flux budget as an aim of their study.
- Most of the other studies follow different aims, some intend to test the suitability of the presented analyzer for EC measurements, others focus on correction factors or error analysis, while some are investigating processes of tropospheric chemistry.

- Many studies do not explicitly give flux uncertainties or detection limits for their systems. In those studies where numbers are mentioned, a large range of values is reported.
- No other EC $\Sigma N_r$ (i.e. total $N_r$) studies were found except for those by authors from this study.

**Changes in the manuscript (1.1):**
We have added the table into the Supplement of the article as some readers may find the information useful. Additional text is added in line 82 to properly link the novelty of the here presented datasets with the literature overview:

"The reader is referred to Table S1 in the supplementary section where an overview of previous eddy-covariance campaigns of different $N_r$ compounds is given. The table highlights the limitations of campaign lengths, a wide range of flux uncertainties, and mainly different aims of previous studies such as testing the suitability of a certain analyzer, investigating processes of tropospheric chemistry, or derivation of correction factors."

**Reviewer comment 1.2:**
*Another example is green-house gas flux; other ESSD papers demonstrated typically for greenhouse gas flux for longer time period (> several ten years), which enables us to discuss the annual carbon budget. Actually you mentioned about the impact of nitrogen deposition on carbon storage in l443-448 in p15, but your WET site dataset for instance is collected for less than 1 yr and could not used for annual deposition/emission. Although this is partially described in Introduction section, more explanation is still required.*

**Authors' response 1.2:**
It is true that we stress the possibility of the presented datasets for investigating potential effects of N deposition on $CO_2$ exchange. We agree that even for the >2.5-yr period at the forest site, it is hardly possible to indicate trends in the respective annual balances, let alone some sort of robust analysis on this time scale would be meaningful. It was meant that continuous $N_r$ flux datasets like the ones that are here presented, offer a useful tool to investigate the relationship between N deposition and $CO_2$ exchange when carried out over multiple years. To clarify this, we have rephrased line 450f. accordingly.

Regarding the length of our presented datasets in comparison to typical greenhouse gas flux times series, it becomes clear that with the above-mentioned reasons such as the fact that campaigns have largely remained experimental due to high costs for devices, maintenance, and operation or the gas phase reactions requiring individual measurements of several species simultaneously (authors' response 1.1, also lines 51-62), the community is just simply not there yet where the $CO_2$ flux community is. It is the aim of this paper to provide a first piece into a direction of more and longer $N_r$ flux measurements and should motivate other groups to work on joint initiatives as well as consolidated longer-term monitoring programs of $N_r$.

**Changes in the manuscript (1.2):**
We have replaced line 446 with the following sentence:

"Finally, continuous $N_r$ flux datasets like the ones presented in this study can be used to analyse interactive effects with other components of biogeochemical cycles when carried out over multiple years."

**Reviewer comment 1.3:**
*- Please add the following data to your repository or if available from other projects (e.g., LTER, fluxnet, ...) please add these link address to your repository website: net radiation, sensible and latent heat, CO2 fluxes over the canopy, soil temp and moisture (for model validation), atmospheric pressure, precipitation, downward long-wave radiation (for model input), plant functional type (PFT), LAI, canopy height, leaf trait, soil type and texture (for input parameters). Management (fertilizer practice) data, too. They are generally required for land surface modeling in addition to your concentration and flux data.*

**Authors' response 1.3:**
We agree that providing the requested data is useful for running land-surface models. We checked the availability and have added those datasets to the repository that we either measured ourselves or where we have access to, see Table R2 for details.

*Table R2: Availability of additional datasets that have been added to the repository website from the peatland (WET) and the forest (FOR) site, where "x" indicates available and "o" indicates that these data will not be added as they were either not recorded or have an insufficient quality due to bad instrument performance. Fertilization is of no relevance at both sites (unmanaged).*

|  | WET site | FOR site |
|---|---|---|
| $R_n$ | x | o |
| $LE$ | x | o |
| $H$ | x | x |
| $F_{CO2}$ | x | (x), only 3 months |
| $T_s$ | x | o |
| SWC | o | o |
| $P_{bar}$ | x | x |
| Precipitation | x (already in initial submission) | x (already in initial submission) |
| PAR | x | o |
| PFT | x | x |
| LAI | o | (x), modeled |
| Canopy height | x | x |
| Soil type | x | x |
| Soil texture | x | x |
| Fertilization | n.a. | n.a. |

**Changes in the manuscript (1.3):**
Additional entries corresponding to the available data listed in the above Table R2 were added to Table 3 in the manuscript. We further added the following lines to Section 8 in line 483f.:

"After the initial submission, some additional site data sheets of interest were added. These can be found in the tables beginning with "additional_WET_site_data" and "additional_FOR_site_data" and comprise $CO_2$, sensible and latent heat fluxes, net radiation, photosynthetically active radiation, soil temperature, barometric pressure, plant functional type, leaf area index, canopy height, soil type, and soil texture, where available."

**Changes in the data repository (1.3):**
The following data tables were added to the data repository:

- additional_FOR_site_data
- additional_FOR_site_characteristics_data
- additional_WET_site_data
- additional_WET_site_characteristics_data

The tables contain the available data outlined in Table R2. Descriptions and units are listed in Table 3 of the manuscript.

**Reviewer comment 1.4:**
*- I think your low-resolution data can be used for validation of your high-resolution data over time series and/or scatter plots. Currently this seems to be used just for dry deposition estimation though.*

**Authors' response 1.4:**
We thank the reviewer for the suggestion and support the idea of verifying the agreement between monthly integrated DELTA (or passive sampler, PS) concentrations and averaged QCL values for the corresponding exposition periods. However, we think that the high-resolution data should be the reference and not the other way around due to precision, accuracy and rigorous calibration of the QCL and no further error sources from post-field wet-chemical lab analyses involved in the DELTA and passive sampler procedures.

For fluxes, a validation of N deposition estimation with a model using monthly concentration input data is not recommended as the time resolution does not resolve micrometeorological variability such as diurnal courses of temperature, radiation, friction velocity or humidity. This is extensively shown and discussed in the publication of Schrader et al. (2018) and would not add any new insight to the here presented dataset publication.

We limit our analysis to the $NH_3$ campaigns at the WET and the FOR site, because for $\Sigma N_r$ at the WET site, DELTA and PS exposition did not overlap with the campaign period of the TRANC and at the FOR site the same topic is discussed in detail in the publications by Wintjen et al. (2022a, 2022b) for $\Sigma N_r$.

Figures R1 to R5 show the respective comparison of QCL vs. low resolution data from DELTA or PS. Data from the latter two low-resolution methods were consistently underestimated compared to QCL values. It must be noted that at the WET site, the data gap in QCL measurements in April 2014 led to a larger difference between QCL and DELTA as the values were generally decreasing from the early April peak due to fertilization on adjacent agricultural sites, hence biasing the QCL average. A full time series would have probably led to a slope much closer to 1 in Figure R2 as was found in the comparison between QCL and DELTA at the FOR site (*cf*. Figure R4).

**Changes in the manuscript (1.4):**
We have added these figures to the supplementary section and extended lines 333f. in Section 5 (Data description) of the manuscript as follows:

"In the following, we briefly highlight key characteristics of the high-resolution TRANC and QCL data. A comparison with low-resolution concentration data from DELTA denuders and passive samplers is given in the supplementary section."

[Figure]

*Figure R1: Concentration time series of NH₃ at the peatland (WET) site. Horizontal red lines correspond to the exposition time of the DELTA denuders. For better comparability, averages of the QCL are shown in blue for the same periods.*

[Figure]

*Figure R2: Scatter plot of NH₃ concentration from QCL and DELTA denuders corresponding to identical periods at the peatland (WET) site.*

[Figure]

*Figure R3: Concentration time series of NH₃ at the forest (FOR) site. Horizontal blue and red lines correspond to the exposition time of the DELTA denuders and passive samplers (PS), respectively. For better comparability, averages of the QCL are shown in black for the same periods.*

[Figure]

*Figure R4: Scatter plot of NH₃ concentration from QCL and DELTA denuders corresponding to identical periods at the forest (FOR) site.*

[Figure]

*Figure R5: Scatter plot of NH$_3$ concentration from QCL and passive samplers (PS) corresponding to the same periods at the forest (FOR) site.*

**Reviewer comment 1.5:**
*- please show the accuracy of gap-filling method using dry deposition models (4.3.2 and 4.4) via comparing the model outputs with original flux data over time series and/or scatter plots. Then you can tune the parameters such as Massad et al. (2010) model to reproduce the original flux data (unnecessary use of default parameter sets, if the results are improved)*

**Authors' response 1.5:**
We appreciate the comment and have added Figures R6 to R8 to the supplementary section. Although the variability in the scatter plots showing valid measured vs. modeled fluxes is relatively high (Figure R6), the impact of the total cumulative exchange appears to be acceptable (Figures R7 and R8). The analysis is limited to the WET NH$_3$ and the FOR TRANC campaigns as no model application was performed for WET TRANC due to missing data of N$_r$ species other than NH$_3$ and due to no flux calculation at FOR NH$_3$ (for reasons see the second part of the response to Comment 1.7).

For the WET NH$_3$ campaign, the model underestimates the QCL measurements by 13.5% when taking all valid measured half-hours into account. At the FOR TRANC campaign, we only show a comparison for the summer months in 2016, where the model overestimates the TRANC measurements by 65%. This is the period of highest fluxes and consequently the highest potential absolute deviations. It is shown in Wintjen et al. (2022b) that over the 2.5-yr campaign the difference between measurements and model is overall in the range of ~25%.

We appreciate the idea of using the valid measured fluxes for fine-tuning parameters in the models. However, as indicated multiple times in Section 6 (Potential applications), the aim of this paper is to present the datasets from the measurements by stressing all required steps for robust data processing and thorough QA/QC. A detailed model parameter investigation would go beyond the scope of this study and comprises one of many potential logical steps for a follow-up paper. We therefore do not intend to include such an analysis in the here presented measurement dataset presentation.

**Changes in the manuscript (1.5):**
Information where the reader is referred to the respective figures have been added after line 298 (end of Section 4.4):

"The accuracy of the gap-filling method from this section and from 4.3.2 by comparing the model outputs with original flux data is shown in the supplementary section."

Figures added to the supplementary section:

[Figure]

*Figure R6: Measured vs. modeled deposition data in half-hourly time resolution at the forest (FOR, left panel) and peatland site (WET, right panel). FOR data comprise the period mid-July to end of September 2016. For the WET site the entire campaign from February to May 2014 is shown.*

[Figure]

*Figure R7: Time series (upper panel) and cumulative curves (lower panel) of measured vs. modeled deposition data in half-hourly time resolution at the peatland site (WET) from February to May 2014.*

[Figure]

*Figure R8: Time series (upper panel) and cumulative curves (lower panel) of measured vs. modeled deposition data in half-hourly time resolution at the forest site (FOR) from mid-July to September 2016.*

**Reviewer comment 1.6:**
*- Table 3 lacks the info about NH3 concentration and flux measurements and low-resolution datasets (while all data are available in the website).*

**Authors' response 1.6:**
We have added the $NH_3$ related entries (concentration, fluxes, gap-filled time series with units and short descriptions) as new lines in Table 3 regardless of their presence in the respective site data table. To clarify this, we added the following sentence to the table caption:

"Note that not every column header exists in every data table due to the focus on the specific gas and gap-filling method."

We have also added the headers from the low-resolution DELTA and passive sampler datasets.

**Changes in the manuscript (1.6):**
Extended caption and additional entries in Table 3 as outlined above.

**Reviewer comment 1.7:**
*- Fig. 3: pls specity if the flux data is gap-filled or not. Also could you add the lower detection limit for the NH3 flux of "n.a." over the forest site?*

**Authors' response 1.7:**
We thank the reviewer for the catch. The information was missing, but is surely important. The idea for the flux data presentation in Figure 3 is to show the most complete time series (column 3) and based on that a robust mean diurnal curve (column 4). Hence, for campaign 1 and 2 at the WET sites, the dataset gap-filled by mean diurnal variation (MDV) and for campaign 3 at the FOR site, the dataset gap-filled by the model is used.

Regarding the lower detection limit, we state in lines 352-354 that the observed low signal-to-noise ratio as a consequence of insufficient high-frequency variability in the low-concentration regime made a robust flux detection impossible and referred to the publication of Zöll et al. (2016).

**Changes in the manuscript (1.7):**
We have added the following information to the figure caption:

"Flux data shown in column 3 are the gap-filled time series using the mean diurnal variation (MDV) method for the WET campaigns and the model-filled time series for the FOR campaign. See Section 4.3 for details. The mean diurnal flux courses in column 4 are based on the given time series in column 3."

For completeness, we have also added the value of the detection limit in the caption of Figure 3:

"Insufficient high-frequency variability in the low-concentration regime of ammonia measured by QCL at the FOR site made a robust covariance detection unreliable with fluxes largely under the lower detection limit of 7.75 ng N $m^{-2}$ $s^{-1}$. $NH_3$ fluxes at the FOR site are therefore not given."

Comment on essd-2021-85
**Anonymous Referee #2**

Referee comment on "Reactive nitrogen fluxes over peatland and forest ecosystems using micrometeorological measurement techniques" by Christian Brümmer et al., Earth Syst. Sci. Data Discuss., https://doi.org/10.5194/essd-2021-85-RC2, 2021

**Reviewer comment 2:**
*This paper describes an important database of nitrogen fluxes over two ecosystems. The site description is good and methodology is presented in details. Essential steps were carried out correctly by the authors, especially regarding the high frequency corrections due to sensor separation and lags between the sonic anemometer and gas analyzers. Gapfilling and uncertainty analysis were also carried out according to current methods, providing estimates of annual depositions.*

**Authors' response 2:**
We appreciate the positive evaluation by the reviewer. We are glad to see that both dataset and text are apparently presented in a clear and understandable way.

Table R1: Overview of literature presenting eddy-covariance measurements of reactive nitrogen compounds. Some additional flux campaigns are listed in the publication of Walker et al. (2020).

| Paper | Compound | Main aim of study | Dataset length | Flux uncertainty / detection limit | Vegetation type |
|---|---|---|---|---|---|
| Ammann et al. (2012) | $\Sigma N_r$ | Suitability of converter for EC measurements | Few weeks are shown for cross-validation with other techniques | ~5 ng N m$^{-2}$ s$^{-1}$ (upper flux detection limit) | Managed grassland |
| Brümmer et al. (2013) | $\Sigma N_r$ | Temporal dynamics, controlling factors, and seasonal N budget | 11 months | ~6.6 ng N m$^{-2}$ s$^{-1}$ (upper flux detection limit) | Cropland (winter wheat) |
| Eugster and Hesterberg (1996) | $NO_2$ | Deriving transfer resistances | Four different periods with a total of 68 days | Not explicitly given | Rural litter meadow |
| Famulari et al. (2004) | $NH_3$ | Suitability of TDLAS system for EC; cross-validation with AGM | 2 months | Not explicitly given, only standard deviation of fluxes for entire campaign | Managed grassland |
| Farmer and Cohen (2008) | $HNO_3$, $\Sigma AN$, $\Sigma PN$ and $NO_2$ | In-canopy chemical analysis | 12 months | Not explicitly given | Ponderosa pine plantation |
| Farmer et al. (2006) | $HNO_3$, $\Sigma AN$, $\Sigma PN$ and $NO_2$ | Suitability of TD-LIF system for EC | 12 months; shorter periods are shown from different seasons | <1 ng N m$^{-2}$ s$^{-1}$; <20% relative errors at low wind speed (<1 m s$^{-1}$) | Ponderosa pine plantation |
| Farmer et al. (2011) | Aerosols ($NH_4$, $SO_4$, $NO_3$) | Suitability of HR-AMS system for EC | 15 days | ~0.4 to 6.4 ng m$^{-2}$ s$^{-1}$ depending on substance and mode; typical single flux measurement was below DL for $NH_4$ fragments | Ponderosa pine plantation |
| Ferrara et al. (2012) | $NH_3$ | Comparison of high-frequency correction methodologies using QC-TILDAS | 13 days | ~75 ng N m$^{-2}$ s$^{-1}$ (flux detection limit) | Cropland (sorghum) |
| Ferrara et al. (2016) | $NH_3$ | Temporal dynamics of $NH_3$ volatilization after slurry application using QC-TILDAS | ~14 days | Only MAE (4700 ng $NH_3$ m$^{-2}$ s$^{-1}$) and RMSE (12000 ng $NH_3$ m$^{-2}$ s$^{-1}$) given | Maize stubbles and Italian ryegrass |

| | | | | | |
|---|---|---|---|---|---|
| Ferrara et al. (2021) | $NH_3$ | Evaluation of measurement errors using QCL spectrometer | 21 days | 13.6 and 20.7 ng m$^{-2}$ s$^{-1}$ at 95 and 99% CI, respectively | Cropland (faba bean) |
| Horii et al. (2004) | NO, $NO_2$, $O_3$ | Impacts of temporal dynamics on tropospheric chemistry and parameterizations | 7 months, but no time series shown | Not explicitly given | Mixed deciduous forest |
| Horii et al. (2006) | $NO_x$, $NO_y$ | Concentration and flux budgets of $N_r$, inferring $HNO_3$, validation of deposition velocities | 5 months, but only time series of ~2 weeks are shown | Not explicitly given | Mixed deciduous forest |
| Marx et al. (2012) | $\Sigma N_r$ | Suitability of converter for capturing all $N_r$ species at high frequency | 1-week validation, 11 months field campaign | Not explicitly given as aim was on concentrations and fast response | Managed grassland and cropland (winter wheat) |
| Min et al. (2014) | NO, $NO_2$ | Comparison of gradient and direct flux measurements; within-canopy chemistry of $NO_x$ | 6 weeks, no time series shown | <8% for NO flux; <6% for $NO_2$ flux; 0.08 ppt m s$^{-1}$ (NO); 0.14 ppt m s$^{-1}$ ($NO_2$) | Ponderosa pine plantation |
| Moravek et al. (2019) | $NH_3$ | Quantify impact of adsorption on time response of the system | 5 months | Median flux detection limit of 2.15 ng m$^{-2}$ s$^{-1}$ | Corn crop field |
| Munger et al. (1996) | $NO_y$, $O_3$ | Response of $NO_y$ deposition to environmental conditions | 5 years | Only given for concentrations (~50 ppt at the mixed forest site and <10 ppt at the spruce woodland) | Mixed deciduous forest and spruce woodland |
| Rummel et al. (2002) | NO | Flux pattern within the canopy | 3 months | 0.07 ng N m$^{-2}$ s$^{-1}$ | Amazonian rain forest |
| Sintermann et al. (2011) | $NH_3$ | Suitability of a CIMS (chemical ionization mass spectrometry) instrument for EC measurements | Few days | 5 ng N m$^{-2}$ s$^{-1}$ | Crop stubble field and cut grassland |
| Sun et al. (2015) | $NH_3$ | Suitability of the open-path $NH_3$ sensor for EC measurements and comparison to other commercial sensors | 2 weeks | 1.3 +/- 0.5 ng m$^{-2}$ s$^{-1}$ | Cattle feedlot |

| Wang et al. (2021) | NH$_3$ | Suitability of the open-path NH$_3$ sensor for EC measurements | 1 week | 7.1 ug N m$^{-2}$ h$^{-1}$ | Subtropical rice paddy |
| Whitehead et al. (2008) | NH$_3$ | Suitability and inter-comparison of different analyzers | 2 campaigns, only few days are presented | Not explicitly given | Managed grassland |

References in Table R1:

Ammann, C., Wolff, V., Marx, O., Brümmer, C., and Neftel, A.: Measuring the biosphere-atmosphere exchange of total reactive nitrogen by eddy covariance, Biogeosciences, 9, 4247−4261, https://doi.org/10.5194/bg-9-4247-2012, 2012.

Brümmer, C., Marx, O., Kutsch, W., Ammann, C., Wolff, V., Flechard, C. R., and Freibauer, A.: Fluxes of total reactive atmospheric nitrogen (ΣNr) using eddy covariance above arable land, Tellus B, 65, 19770, doi:10.3402/tellusb.v65i0.19770, 2013.

Eugster, W. and Hesterberg, R.: Transfer resistances of NO2 derived from eddy correlation flux measurements over a litter meadow at a rural site on the Swiss Plateau. Atmos. Environ., 30, 1247−1254, 1996.

Famulari, D., Fowler, D., Hargreaves, K., Milford, C., Nemitz, E., Sutton, M. A., and Weston, K.: Measuring eddy covariance fluxes of ammonia using tunable diode laser absorption spectroscopy, Water, Air Soil Pollut. Focus, 4, 151−158, 2004.

Farmer, D. K. and Cohen, R. C.: Observations of HNO3, ΣAN, ΣPN and NO2 fluxes: evidence for rapid HOx chemistry within a pine forest canopy. Atmos. Chem. Phys., 8(14), 3899−3917, doi:10.5194/acp-8-3899-2008, 2008.

Farmer, D. K., Wooldridge, P. J., and Cohen, R. C.: Application of thermal-dissociation laser induced fluorescence (TD-LIF) to measurement of HNO3, Σalkyl nitrates, Σperoxy nitrates, and NO2 fluxes using eddy covariance, Atmos. Chem. Phys., 6, 3471–3486, https://doi.org/10.5194/acp-6-3471-2006, 2006.

Farmer, D. K., Kimmel, J. R., Phillips, G., Docherty, K. S., Worsnop, D. R., Sueper, D., Nemitz, E., and Jimenez, J. L.: Eddy covariance measurements with high-resolution time-of-flight aerosol mass spectrometry: a new approach to chemically resolved aerosol fluxes, Atmos. Meas. Tech., 4, 1275–1289, https://doi.org/10.5194/amt-4-1275-2011, 2011.

Ferrara, R. M., Loubet, B., Di Tommasi, P., Bertolini, T., Magliulo, V., Cellier, P., Eugster, W., and Rana, G.: Eddy covariance measurement of ammonia fluxes: Comparison of high frequency correction methodologies, Agr. Forest Meteorol., 158–159, 30–42, 2012.

Ferrara, R. M., Carozzi, M., Di Tommasi, P., Nelson, D. D., Fratini, G., Bertolini, T., Magliulo, V., Acutis, M., and Rana, G.: Dynamics of ammonia volatilisation measured by eddy covariance during slurry spreading in north Italy. Agric. Ecosys. Environ., 219, 1–13, https://doi.org/10.1016/j.agee.2015.12.002, 2016.

Ferrara, R.M., Di Tommasi, P., Famulari, D., and Rana, G.: Limitations of an Eddy-Covariance System in Measuring Low Ammonia Fluxes. Boundary-Layer Meteorol 180, 173–186, https://doi.org/10.1007/s10546-021-00612-6, 2021.

Horii, C. V., Munger, J. W., and Wofsy, S. C.: Fluxes of nitrogen oxides over a temperate deciduous forest, J. Geophys. Res., 109, D08305, doi:10.1029/2003JD004326, 2004.

Horii, C. V., Munger, J. W., Wofsy, S. C., Zahniser, M., Nelson, D., and McManus, J. B.: Atmospheric reactive nitrogen concentration and flux budgets at a Northwestern US forest, Agr. Forest Meteorol., 136, 159–174, 2006.

Marx, O., Brümmer, C., Ammann, C., Wolff, V., and Freibauer, A.: TRANC – a novel fast-response converter to measure total reactive atmospheric nitrogen, Atmos. Meas. Tech., 5, 1045–1057, https://doi.org/10.5194/amt-5-1045-2012, 2012.

Min, K.-E., Pusede, S. E., Browne, E. C., LaFranchi, B. W., and Cohen, R. C.: Eddy covariance fluxes and vertical concentration gradient measurements of NO and NO2 over a ponderosa pine ecosystem: observational evidence for within-canopy chemical removal of NOx, Atmos. Chem. Phys., 14, 5495–5512, https://doi.org/10.5194/acp-14-5495-2014, 2014.

Moravek, A., Singh, S., Pattey, E., Pelletier, L., and Murphy, J. G.: Measurements and quality control of ammonia eddy covariance fluxes: A new strategy for high frequency attenuation correction, Atmospheric Measurement Techniques, 12, 6059–6078, https://doi.org/10.5194/amt-12-6059-2019, 2019.

Munger, J. W., Wofsy, S. C., Bakwin, P. S., Fan, S.-M., Goulden, M. L., Daube, B. C., Goldstein, A. H., Moore, K. E., and Fitzjarrald, D. R.: Atmospheric deposition of reactive nitrogen oxides and ozone in a temperate deciduous forest and a subarctic woodland: 1. Measurements and mechanisms, J. Geophys. Res., 101(D7), 12639– 12657, doi:10.1029/96JD00230, 1996.

Rummel, U., Ammann, C., Gut, A., Meixner, F. X., and Andreae, M. O.: Eddy covariance measurements of nitric oxide flux within an Amazonian rain forest, J. Geophys. Res., 107, 8050, doi:10.1029/2001JD000520, 2002.

Sintermann, J., Spirig, C., Jordan, A., Kuhn, U., Ammann, C., and Neftel, A.: Eddy covariance flux measurements of ammonia by high temperature chemical ionisation mass spectrometry, Atmos. Meas. Tech., 4, 599–616, doi:10.5194/amt-4-599-2011, 2011.

Sun, K., Tao, L., Miller, D. J., Zondlo, M. A., Shonkwiler, K. B., Nash, C., and Ham, J. M.: Open-path eddy covariance measurements of ammonia fluxes from a beef cattle feedlot, Agr. Forest Meteorol, 213, 193−202, https://doi.org/10.1016/j.agrformet.2015.06.007, 2015.

Walker, J. T., Beachley, G., Zhang, L., Benedict, K. B., Sive, B. C., and Schwede, D. B.: A review of measurements of air-surface exchange of reactive nitrogen in natural ecosystems across North America, Science of The Total Environment, 698, 133975, https://doi.org/10.1016/j.scitotenv.2019.133975, 2020.

Wang, K., Kang, P., LU, Y., Zheng, X., Liu, M., Lin, T.-J., Butterbach-Bahl, K., and Wang, Y.: An open-path ammonia analyzer for eddy covariance flux measurement, Agr. Forest Meteorol, 308−309, 108570, https://doi.org/10.1016/j.agrformet.2021.108570, 2021.

Whitehead, J. D., Twigg, M., Famulari, D., Nemitz., E., Sutton, M. A., Gallagher, M. W., and Fowler, D.: Evaluation of Laser Absorption Spectroscopic Techniques for Eddy Covariance Flux Measurements of Ammonia, Environ. Sci. Technol., 42, 2041−2046, 2008.

References from response document:

Schrader, F., Schaap, M., Zöll, U., Kranenburg, R., and Brümmer, C.: The hidden cost of using low-resolution concentration data in the estimation of NH3 dry deposition fluxes. Scientific Reports, 8(1), 969, https://doi.org/10.1038/s4159 8-017-18021-6, 2018.

Wintjen, P., Schrader, F., Schaap, M., Beudert, B., and Brümmer, C.: Forest-atmosphere exchange of reactive nitrogen in a remote region – Part I: Measuring temporal dynamics. Biogeosciences, *in press*, 2022a.

Wintjen, P., Schrader, F., Schaap, M., Beudert, B., and Brümmer, C.: Forest-atmosphere exchange of reactive nitrogen in a remote region – Part II: Modeling annual budgets. Biogeosciences, *under review*, 2022b.